# Flash healing of laser-induced graphene

Le Cheng[1,2], Chi Shun Yeung[1,2], Libei Huang[1,3], Ge Ye[1], Jie Yan[4], Wanpeng Li [4], Chunki Yiu[5], Fu-Rong Chen [4], Hanchen Shen[6], Ben Zhong Tang [6,7], Yang Ren [8,9], Xinge Yu [5] ✉ & Ruquan Ye [1,2] ✉

The advancement of laser-induced graphene (LIG) technology has streamlined the fabrications of flexible graphene devices. However, the ultrafast kinetics triggered by laser irradiation generates intrinsic amorphous characteristics, leading to high resistivity and compromised performance in electronic devices. Healing graphene defects in specific patterns is technologically challenging by conventional methods. Herein, we report the rapid rectification of LIG's topological defects by flash Joule heating in milliseconds (referred to as F-LIG), whilst preserving its overall structure and porosity. The F-LIG exhibits a decreased $I_D/I_G$ ratio from 0.84 – 0.33 and increased crystalline domain from Raman analysis, coupled with a 5-fold surge in conductivity. Pair distribution function and atomic-resolution imaging delineate a broader-range order of F-LIG with a shorter C-C bond of 1.425 Å. The improved crystallinity and conductivity of F-LIG with excellent flexibility enables its utilization in high-performance soft electronics and low-voltage disinfections. Notably, our F-LIG/polydimethylsiloxane strain sensor exhibits a gauge factor of 129.3 within 10% strain, which outperforms pristine LIG by 800%, showcasing significant potential for human-machine interfaces.

Graphene is a widely renowned two-dimensional material for its remarkable mechanical, electrical, and thermal properties[1,2]. This atom-thin material, prepared by the mechanical exfoliation of natural graphite using adhesive tape, was shown to have high electron mobility in ambient conditions in 2004[3]. Since then, various preparation techniques have been developed to facilitate its academic research and industrial applications. The most commonly employed strategies include chemical vapor deposition[4,5], liquid-phase exfoliation of graphite[6,7], and reduction of graphene oxide[8,9]. These early production methods, however, have notable limitations that hinder their use in commercial and large-scale manufacturing, such as the requirement

for specialized equipment, the use of harsh oxidizing and reducing agents, high energy consumption, and labor-intensive processes.

In 2014, a significant advancement in graphene synthesis was achieved with the development of laser-induced graphene (LIG)[10,11]. It is a bottom-up approach that directly converts carbonous precursors, including natural and synthetic polymers[12,13], and small molecules[14,15], into three-dimensional (3D) porous graphene through laser irradiation. This technology enables the convenient and scalable production of LIG using a commercial laser cutting machine in ambient atmosphere, offering several advantages over conventional graphene, such as high porosity, good flexibility, cost-effectiveness, and mask-free

[1]Department of Chemistry, State Key Laboratory of Marine Pollution, City University of Hong Kong, Hong Kong 999077, P. R. China. [2]City University of Hong Kong Research Institute, Shenzhen, Guangdong 518057, P. R. China. [3]Division of Science, Engineering and Health Study, School of Professional Education and Executive Development (PolyU SPEED), The Hong Kong Polytechnic University, Hong Kong 999077, P. R. China. [4]Department of Materials Science and Engineering, Time-resolved Aberration Corrected Environmental Electron Microscope Unit, City University of Hong Kong, Hong Kong 999077, P. R. China. [5]Department of Biomedical Engineering, City University of Hong Kong, Hong Kong 999077, P. R. China. [6]School of Science and Engineering, Shenzhen Institute of Aggregate Science and Technology, The Chinese University of Hong Kong, Shenzhen (CUHK-Shenzhen), Guangdong 518172, P. R. China. [7]Department of Chemistry, Hong Kong Branch of Chinese National Engineering Research Center for Tissue Restoration and Reconstruction, State Key Laboratory of Molecular Neuroscience, The Hong Kong University of Science and Technology, Hong Kong 999077, P. R. China. [8]Department of Physics, City University of Hong Kong, Hong Kong 999077, P. R. China. [9]Centre for Neutron Scattering, City University of Hong Kong, Kowloon, Hong Kong 999077, P. R. China. ✉e-mail: xingeyu@cityu.edu.hk; ruquanye@cityu.edu.hk

patterning. The versatility and adaptability of the LIG technique have garnered considerable interest across multiple disciplines, such as electronics[16,17], catalysis[18,19], and sterilization[20]. Since its discovery, a wide range of laser sources including infrared (IR)[10,21], visible[22–24], and ultraviolet (UV)[25,26], have been employed for LIG fabrication. However, it is important to note that these laser pulses have notably short durations, ranging from microseconds to femtoseconds[12,25,27–29], which leads to ultrafast kinetics that predominantly yield amorphous structures in LIG. While this polycrystalline structure has advantages for energy storage and electrocatalysis due to its rich electron states near the Fermi level and abundant active sites for substrate binding[10,14,18], it can also result in poor electrical conductivity, thereby limiting its performance in electronic devices such as electrical heaters and sensors. Therefore, it is crucial to control the defect density and optimize the crystal structure in LIG for tailored applications.

Current methods for healing defects in graphene materials typically involve bulk heating and chemical reduction. These approaches have shown success in preparing graphene films, powders, and colloids, but are not directly applicable for healing defects in LIG patterns. For instance, high-quality reduced graphene oxide with high conductivity is often obtained through furnace annealing at high temperatures, ranging from several hundreds to thousands of degrees Celsius, under inert or reductive atmospheres[30,31]. However, this process can cause the carbonization of the entire material, leading to the ablation of polymer substrates thus damage the LIG patterns. Chemical reductants such as hydrazine[32], sodium borohydride[33], and hydrohalic acids[34], are capable of reacting with oxygen-containing functional groups like epoxy and carboxyl groups. However, chemical reduction is incapable of inducing atom rearrangements to transform amorphous carbon rings into crystalline structures.

In this work, we demonstrate the effectiveness of flash Joule heating (FJH) in healing defects present in LIG at the atomic level, while preserving its macroscopic geometry and microscale 3D structure. FJH involves the passage of high direct current (DC) pulses through conductive materials, which enables rapid and intense resistive heating and has found recent applications in the fabrication and processing of carbon materials. For example, Hu's group used FJH to covalently weld carbon nanofibers and enhance graphitization, obtaining a highly conductive carbon network[35]. Tour's group employed FJH to achieve the rapid and large-scale production of graphene from diverse feedstocks such as carbon black, coal, petroleum coke, waste foods, and plastics[36,37]. By controlling the FJH process of carbon nanotubes, hybrids with tunable carbon nanotube/graphene ratio could be obtained[38]. These reports primarily focus on the production of carbon allotropes and lack control over the patterning of the materials. Compared with traditional furnace annealing, FJH offers distinct advantages as it is fast and localized. The heating duration of FJH is relatively short, typically lasting from tens of milliseconds to seconds. However, it is essential to note that the electrical pulses used in FJH are still significantly longer than the laser pulses employed in LIG fabrication. The longer heating durations of FJH may facilitate the reorganization of carbon atoms within LIG, leading to improved order. Inspired by the pioneering works and based on the above analysis, we believe that FJH holds promise for rectifying topological defects present in LIG patterns. Herein, by applying a brief high-power DC pulse to the pre-formed LIG conductive patterns, we achieve the facile and straightforward fabrication of flexible and porous graphene patterns with high crystallinity, referred to as flash Joule heated LIG (F-LIG). Raman spectrometry, together with advanced atomic imaging techniques and X-ray pair distribution functions, reveal the reduction of defect density and the improvement of crystalline size in F-LIG. This improvement in structural characteristics leads to a notable reduction in resistivity by up to 80% compared to unmodified LIG, which opens up opportunities for its utilization in high-performance electronic devices. As a proof of concept, F-LIG is utilized as sensing components

in strain sensors, achieving a substantial improvement in sensitivity and showcasing significant potential in various applications, such as subtle movement detection, information encryption and transmission, and human-machine interfaces. Additionally, we also reveal an enhanced bacterial killing property of F-LIG under low voltages, presenting new possibilities for the development of antibacterial surfaces using LIG in healthcare settings and public spaces.

## Results and Discussion

### F-LIG fabrication and FJH progress investigations

The preparation process for F-LIG is shown in the schematic diagram in Fig. 1a. This two-step procedure involves the scribing of original LIG patterns onto a PI film under ambient conditions, followed by the FJH process. In the FJH procedure, a high DC voltage is applied for an ultrashort duration to the pre-formed LIG patterns within a vacuum chamber. In order to ensure consistent heating area, LIG are patterned into dumbbell shapes (Supplementary Fig. 1). This design allows the FJH to occur primarily at the thinner middle section due to its higher resistance. Silver paste is applied to the wider ends to establish reliable electrical contact (Supplementary Fig. 2). First, different voltages with a pulse duration of 20 ms are applied to the LIG patterns with dimensions of 1 mm × 10 mm, and the resultant sample is denoted as F-LIG-FJH voltage. Figure 1b illustrates the observation of the dazzling flash of black-body radiation emitted from the LIG patterns, confirming the successful execution of the FJH process. This characteristic phenomenon signifies the instantaneous generation of heat resulting from the flow of current. Different flash intensities corresponding to the applied FJH voltages ranging from 150 – 190 V, demonstrate different temperatures reached. The LIG patterns emit an orange-red glow at potentials of 150 and 160 V and the flash emission color turns to white as the voltage rises to 180 and 190 V, which indicates the production of significant thermal energy. As shown in Fig. 1c, the temperatures achieved under each applied voltage (150-190 V) are measured to be around 1300, 1700, 2100, 2300, and 2500 °C, respectively. The resistance of the LIG patterns was measured before and after FJH treatment to investigate the impact of FJH on the electrical properties of LIG. As presented in Fig. 1d, initially, the LIG patterns exhibits an average resistance of around 590 Ω. Upon subjecting to the FJH treatment, a noticeable reduction in resistance is observed and the resistance shows a clear decreasing trend with increasing FJH voltages. Notably, the resistance of F-LIG-190 V decreases significantly to approximately 120 Ω, corresponding to a remarkable 5-fold increase in conductivity.

To provide further insights into the relationship between the applied voltage and the FJH process, instantaneous voltage ($U$) and current ($I$) were recorded during the FJH procedure. Based on these measurements, we derived the instantaneous areal power density ($P_A$) and areal energy density ($E_A$) using the equations $P_A = UI/A$ and $E_A = UI/A\,dt$, respectively. Here $A$ represents the area of the LIG patterns, and $t$ denotes the discharging time. As shown in Fig. 1e-g, the voltage is programmed with a switch-on duration of 20 ms. Under each applied voltage, the resulting current and power density exhibit similar profiles. As the applied voltage increases, the integrated $E_A$, i.e., the generated Joule heat, as well as the temperature reached, show an increasing trend (Fig. 1c). Specifically, under 150 and 160 V (corresponding to an $E_A$ value of 10.5 and 13.4 J cm$^{-2}$, respectively), the current and power density are proportional and exhibit similar profile to the voltage. As voltages increased to 170 V ($E_A \sim 17.6$ J cm$^{-2}$) or higher, the current and power density curves initially follow a similar tread as the voltage but then experience a sudden surge, indicating a notable reduction in the resistance of the LIG patterns. When the FJH voltage is 190 V, there appears a maximum improvement in electrical conductivity. In this case, $P_A$ and corresponding $E_A$ reach ~2100 W cm$^{-2}$ and 27.55 J cm$^{-2}$, respectively, indicating rapid and significant energy input and heat generation within the LIG patterns. Additionally, it is

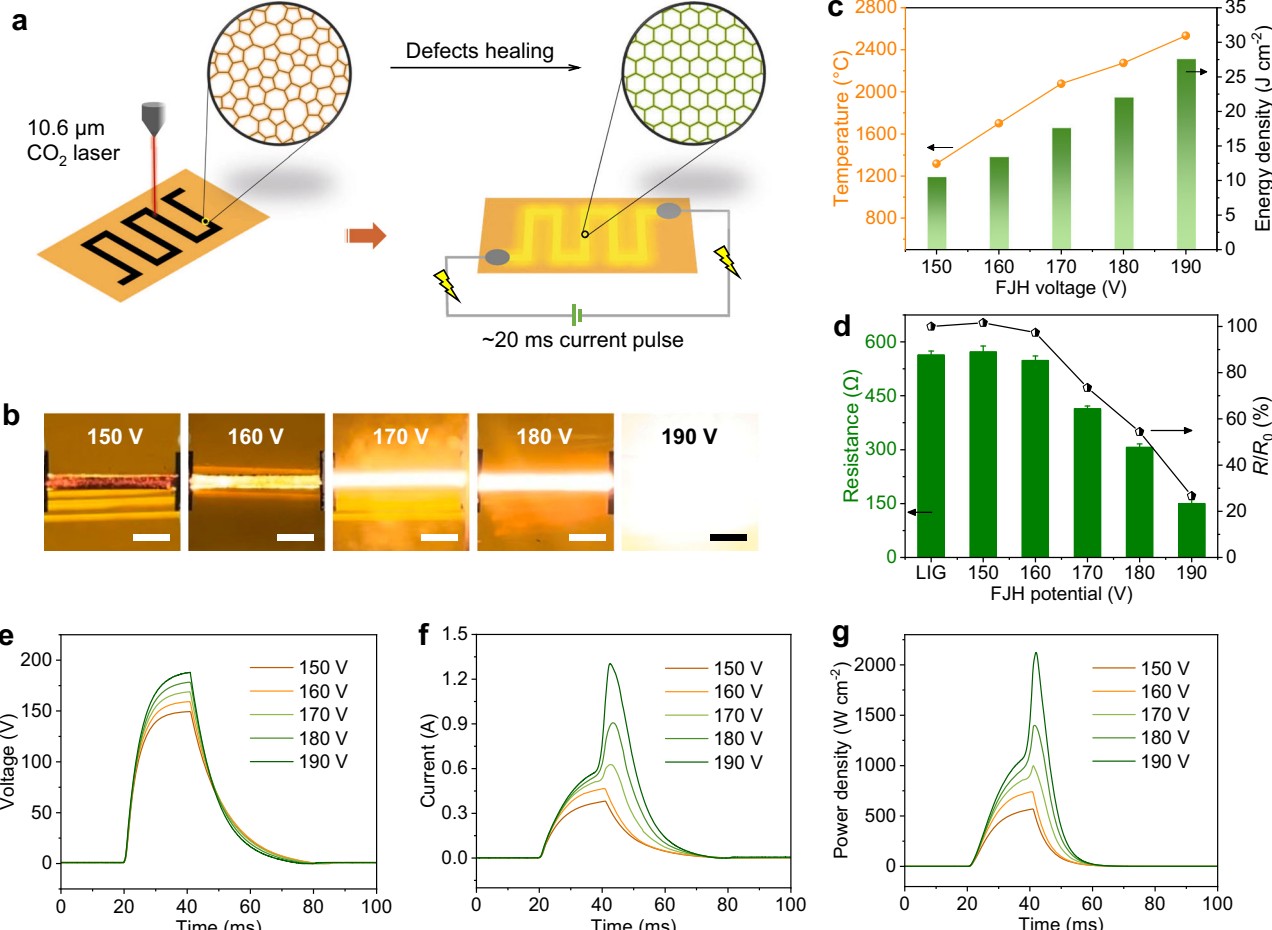

**Fig. 1 | Fabrication of flash Joule heated laser-induced graphene (F-LIG) and investigation of the flash Joule heating (FJH) progress. a** Schematic diagram of the F-LIG fabrication. **b** Digital photograph of LIG patterns (1 mm × 10 mm) during the FJH process under different voltages with a pulse duration of 20 ms. Scale bars are 3 mm. **c** Temperature reached and areal energy density during the FJH process. **d** Resistance and the resistance reduction ratio ($R/R_0$) of F-LIG samples compared to original LIG. Error bars represent the standard deviation of three independent measurements. **e** Voltage, (**f**) current, and (**g**) areal power density profiles during the FJH process.

observed that the onset of current surge happens earlier with the FJH voltage. These observations suggest the presence of a threshold for the generated Joule heat or temperature to trigger the flash healing effect. Below this threshold, corresponding to the cases of 150 and 160 V, the LIG patterns can only be heated to moderate temperatures (below 1700 °C), which is insufficient for the arrangement of carbon atoms. On the other hand, excessively high voltage, for example, 200 V, leads to excessive transient heating and gas release, resulting in the breakdown of the LIG patterns (Supplementary Fig. 3).

## Structural characterization of F-LIG

Raman spectroscopy was employed to examine the defect and layer structure of the LIG and F-LIG samples. As depicted in Fig. 2a, the Raman spectrum of original LIG exhibits prominent D band and G band, which arise from the breathing mode of $sp^2$ carbon atoms in distorted lattice and the in-plane vibration of $sp^2$ carbon atoms in hexagonal lattice, respectively[39,40]. Moreover, a blunt 2D band is observed, indicating the presence of multiple layers in LIG[41]. After FJH treatment, the Raman spectra of F-LIG samples exhibit a significant decrease in the D band with increasing applied voltage, which is indicative of a reduction in density of defects, such as vacancies and disorder. To enable more detailed quantitative analysis, the $I_D/I_G$ ratio was calculated. This ratio serves as an indicator of the degree of defects present in graphitic materials. Additionally, the $I_D/I_G$ ratio can be used to determine the crystalline size ($L_a$) along the a-axis through the

equation $L_a = (2.4 \times 10^{-10}) \times \lambda_l^4 \times (I_G/I_D)$, where $\lambda_l$ is the wavelength of the Raman laser (532 nm)[42]. As shown in Fig. 2c, the $I_D/I_G$ of original LIG is 0.84, corresponding to an $L_a$ of 22.9 nm. In comparison, F-LIG-190 V displays a significantly reduced $I_D/I_G$ of 0.33, and the corresponding $L_a$ reaches ~60 nm, which is ~2.6 times larger than that in LIG. These results imply that the FJH treatment can considerably promote the healing of structural defects in graphene, resulting in the growth and enlargement of the crystalline domains. In addition to the decline of D band, it can be observed that the 2D band exhibits noticeable sharpening and narrowing as the FJH voltage increased. Figure 2d presents the full width at half maximum (FWHM) of the 2D band and the $I_{2D}/I_G$ ratio, which are regarded to be associate with the structural quality and layer count of graphene[41,43]. With the increase of applied voltage, the FWHM of the 2D band dropped from 109.4 cm$^{-1}$ for LIG to 63.8 cm$^{-1}$ for F-LIG-190V. Simultaneously, the $I_{2D}/I_G$ increased from 0.73 – 1.05. The narrower 2D band FWHM and higher $I_{2D}/I_G$ ratio suggests the improved quality and reduced stacks. One possible explanation for the reduced stacking is that the instantaneous heat generated by FJH treatment helps the expansion of graphene layers. A closer look reveals that the 2D peak in F-LIG samples exhibit blue shift with the increase of FJH voltage, as depicted in the enlarged spectra in Fig. 2b. A maximum shift of ~27 cm$^{-1}$ occurs when the applied voltage is 190 V. This shift may be attributed to several possible factors, including changes in the graphene layer count, induced strain in the lattice, and modifications in the electronic band structure[44,45].

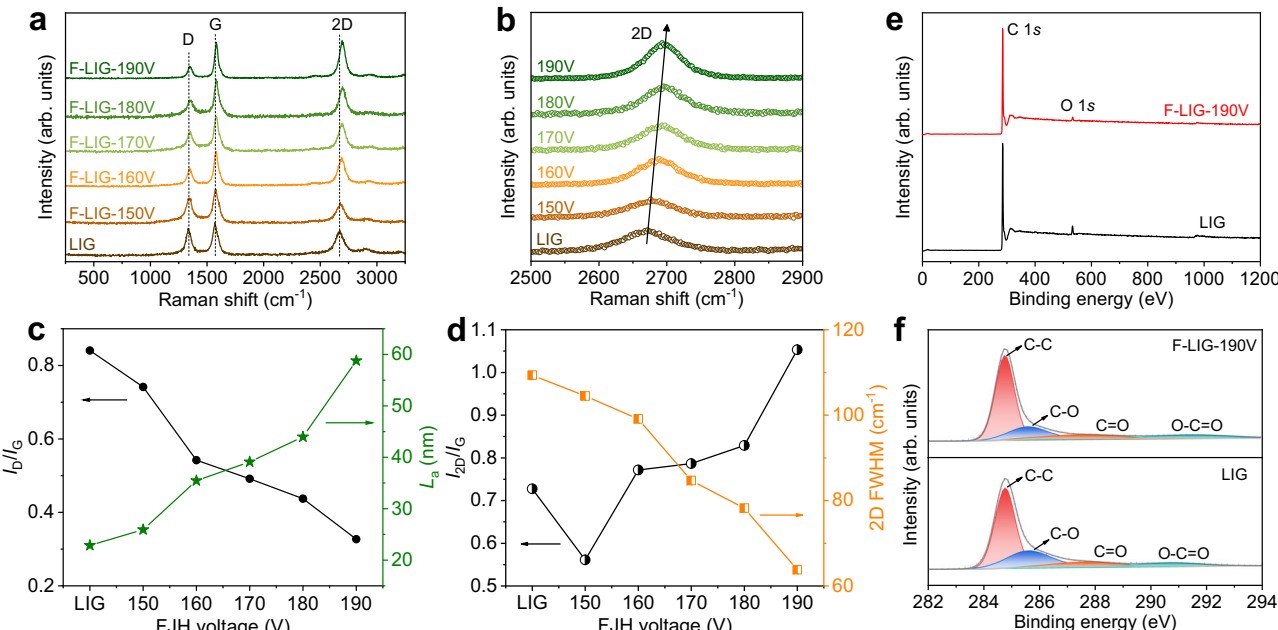

**Fig. 2 | Investigation of the defect structure and atomic binding state. a** Raman spectrum of LIG and F-LIG samples. The vertical dashed lines indicate the positions of the D band, G band, and 2D band of LIG. **b** is the enlarged view of the 2D band region. The arrow indicates the blue shift of the 2D band with the FJH voltage. **c** $I_D/I_G$ and corresponding $L_a$, **d** $I_{2D}/I_G$ and 2D band FWHM of LIG and F-LIG samples. **e** X-ray photoelectron spectroscopy (XPS) survey spectra and (**f**) C 1 $s$ spectra of LIG and F-LIG-190 V. The black solid lines are the raw profiles, and the shaded curves represent carbon species with different atomic binding states.

The elements content and atomic binding state of LIG and F-LIG were analyzed by X-ray photoelectron spectroscopy (XPS). As shown in the survey spectra (Fig. 2e), both LIG and F-LIG exhibits a prominent C 1 $s$ peak at 284 eV, along with a weak O 1 $s$ peak at 530 eV[19]. The initial LIG possesses a high carbon content of 96.28% and relatively low oxygen content (2.69%) and nitrogen content (1.03%). After the FJH treatment, the carbon content increases further to 98.53%, while the oxygen and nitrogen content are reduced to 0.78% and 0.69%, respectively. Further analysis of the deconvoluted C 1 $s$ spectrum (Fig. 2f) reveals that the decrease in oxygen content is primarily attributed to the reduction of C-O bonds. These results indicate that the FJH treatment effectively removes oxygen- and nitrogen-containing groups, which is beneficial for the preparation of graphene material with higher purity and fewer defects.

According to the equation $E_A = UI/A\,dt$, energy density of the FJH process is also proportional to the discharging time. Thus, we conducted additional investigations to assess the influence of pulse durations on the FJH process. Constant voltage of 130 V was applied to the LIG patterns (1 mm × 10 mm) for various durations (20, 30, 40, 50 and 60 ms). As shown in Supplementary Fig. 4, when the pulse duration is 20 and 30 ms, the current exhibits similar profiles with voltage. Correspondingly, there is minimal change in resistance after treatment (Supplementary Fig. 5). As the duration prolongs, an abrupt increase in current is observed, indicating a decrease in resistance caused by the high temperature-induced defect healing. At a pulse duration of 40 ms, the resistance exhibits a slight decrease from 600 Ω to 530 Ω. This resistance is further reduced to 150 Ω when the pulse duration increases to 50 ms. Besides, the D band intensity in the Raman spectra also starts to decrease when the pulse duration is 40 ms and becomes significantly lower after the 50 ms FJH treatment. However, further extending heating time to 60 ms leads to the breakage of the LIG layer, as reflected by the sudden decrease in current. It should be noted that, with the same voltage increments, shorter pulse durations result in smaller increments in energy density, allowing for finer control over the FJH process. Moreover, shorter heating time also minimizes heat dissipation. Based on these considerations, the lower limit of the pulse

duration (20 ms) achievable by the power supply is selected unless otherwise specified.

In order to investigate the impact of FJH on the microstructure of LIG, scanning electron microscopy (SEM) images were collected (Supplementary Fig. 6). The original LIG sample demonstrates a foam-like structure with discernible laser scribing paths in horizontal direction. An enlarged view reveals the existence of fluffy fibrous structures on the surface and spherical micropores beneath. This high-porosity structure is generated due to the release of gaseous matter during the laser manufacturing process. After the FJH treatment, the sponge-like structure of LIG is well-preserved. The 3D porous networks remain continuous, and the micropores are maintained. As seen in the enlarged images, with the increase in FJH voltage, the F-LIG samples exhibit slightly larger micropore size compared to the original LIG. This observation is likely due to the release of gases during the FJH process.

## Atomic-scale morphology study

The atomic-scale morphology of LIG before and after FJH treatment was further examined using high-resolution transmission electron microscopy (HRTEM). As shown in Fig. 3a, for the graphene sheets in original LIG, a prominent characteristic is the disordered arrangement of atoms, which includes a significant presence of pentagons-heptagons pairs (Fig. 3b). This disorder is due to the amorphous nature induced by the rapid heating and cooling during laser irradiation in the microsecond timescale. In contract, graphene sheets in F-LIG-190 V exhibit a wide range of highly ordered hexagonal carbon lattice (Fig. 3c and d). This observation is consistent with the decreased defect concentrations and increased crystallinity revealed by the Raman analysis. These findings further confirm the amorphous-to-crystalline transformation achieved by the FJH technique.

Pair distribution functions (PDFs) analysis was employed to gain insight into the interatomic distance of LIG and F-LIG. As shown in Fig. 3e, all the samples display the first two primary bands corresponding to the nearest planar pair distances at around 1.4 Å and 2.5 Å, consistent with previous literature[14,18]. Fig. 3f presents an enlarged view

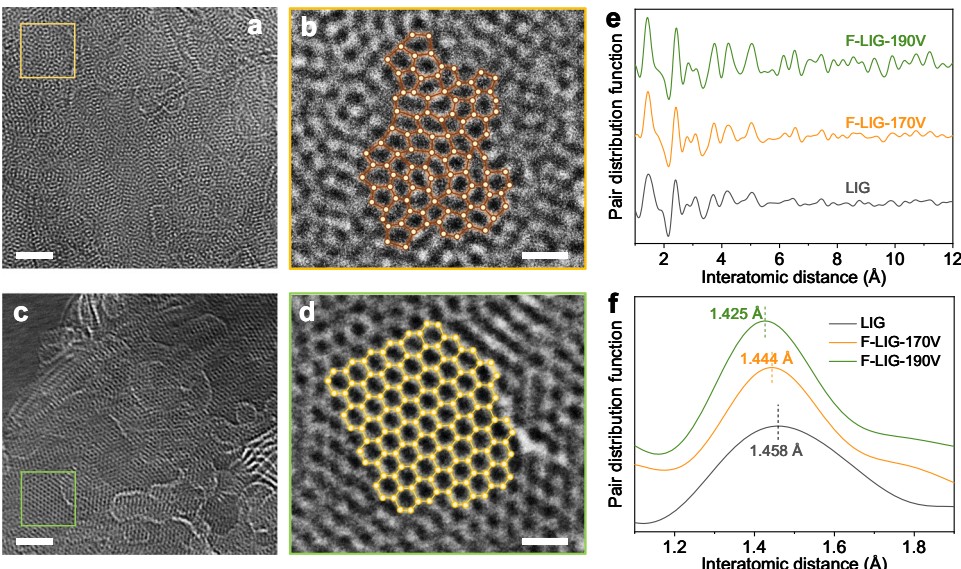

**Fig. 3 | Investigation of topological structure at atomic level.** High-resolution transmission electron microscopy (HRTEM) images of (**a**) LIG and (**c**) F-LIG-190 V. Scale bars are 5 nm. **b** and (**d**) are the enlarged view of the selected area in (**a**) and (**c**) respectively. The atomic ring structures are highlighted. Scale bars are 0.5 nm. **e** Pair distribution functions (PDFs) of LIG and F-LIG samples treated under different voltages. **f** is the enlarged view of the region in (**e**) from 1.1 – 1.9 Å.

of the range 1.1–1.9 Å to further explore the change of interatomic distances. For LIG, the nearest-neighbor band is centered at 1.458 Å, with a broad discrete range. After FJH treatment, this band becomes noticeably narrower and shifts to shorter distance of 1.444 Å and 1.425 Å for F-LIG-170 V and F-LIG-190 V, respectively. The more concentrated distribution of nearest-neighbor bonds, along with the reduction in bond distances, suggests a structural transformation from a five- and seven-membered ring predominant configuration to one with a higher abundance of six-membered rings. This indicates an enhancement in the hexagonal graphitic structure of the treated F-LIG samples. Besides, the subsequent bands after the second nearest neighbors also become more pronounced in F-LIG compared to LIG. This observation suggests the presence of more long-range ordered structures in F-LIG. Taken together, these above results provide compelling evidence of the FJH-induced amorphous-to-crystalline structural transformation at the atomic scale and from a statistical perspective.

**Adaptability of FJH technique to LIG materials**
There are earlier studies mentioning that multiple laser passes can enhance the conductivity of LIG by promoting graphitization[46–48]. Our investigation demonstrates that the reduction in D band intensity and resistance is primarily observed after the second lasing pass (Supplementary Fig. 7). Subsequent lasing cycles exhibit only slight additional reduction in these properties. The resistance reduction is also modest (from ~600 Ω – ~350 Ω, LIG dimension: 1 mm × 10 mm). These results are consistent with previous literature reports[46]. Furthermore, it is noticed that repeated lasing cycles leads to the collapse of LIG (Supplementary Fig. 8). This damage will destructively impact the structural integrity and practical applications. In comparison, it is worth noting that FJH treatment is capable to achieve significant improvement in conductivity, and at the same time, the LIG patterns retains their original appearance, displaying no cracks or deformation (Supplementary Fig. 9). This is because FJH treatment does not subject the material to prolonged exposure of high temperatures or uneven heating, minimizing the potential of structural damage. The retained structural integrity of the resultant F-LIG samples ensures the stability and reliability for subsequent applications and devices. Moreover, the adaptability of the FJH treatment is demonstrated through the successful application on

LIG patterns with varying shapes and dimensions, as well as different substrates including polyethersulfone (PES), poly(ether-ether-ketone) (PEEK), and PES/lignin. The triggering of the FJH process is evident from the observation of a bright flash. Subsequently, the resulting F-LIG exhibit reduced resistance and improved Raman signals (Supplementary Figs. 10–13). These expansions highlight the significant implications and substantial potential of this technique in a wider range of applications. Furthermore, we demonstrated that the defect healing effect of the FJH technique can not only work with the LIG patterns but also with LIG in powder form. To achieve this, powdered LIG was scraped off and subjected to FJH in a quartz tube (Supplementary Fig. 14). In this experimental setup, the FJH process was triggered by applying DC voltages of 70 V, 100 V, and 130 V for approximately 100 ms. Similarly, an intense flash generated from rapid heat accumulation is observed during the process (Supplementary Fig. 15). Furthermore, compared with the LIG patterns, the Raman spectra of powder sample show a similar trend of variation (Supplementary Fig. 16). As the applied voltage increases, the $I_D/I_G$ decreases from 0.84 – 0.27, and the $L_a$ increases from 23.0 – 71.9 nm. The identical change degree in these parameters between the patterned and powdered LIG suggests that both samples undergo a comparable level of defect healing. Additionally, X-ray powder diffraction (XRD) patterns were collected to reveal the revolution in crystallinity in F-LIG (Supplementary Fig. 17). Across all the samples, the characteristic peak centered at $2\theta \approx 26°$, assigned to the (002) facet of graphene, is observed. The initial LIG exhibits a broad peak locates precisely at $2\theta = 25.9°$, corresponding to an interlayer spacing ($I_c$) of ~3.47 Å. After FJH treatment, this characteristic peak gradually increases in intensity and shifts to $2\theta = 26.4°$, corresponding to a reduced interlayer spacing of 3.41 Å. These two $I_c$ values are consistent with the reported literature values for LIG and flash graphene[10,36]. These observations point to the transformation of LIG from an amorphous to a crystalline phase, accompanied by a reduction in the interlayer spacing between the graphene sheets. This reduction could be attributed to the elimination of functional groups or impurities between adjacent layers. Given the comparable level of defect healing observed in both the powered and patterned LIG samples, it can be inferred that the XRD analysis based on the powder sample holds true to some extent for the patterned sample as well.

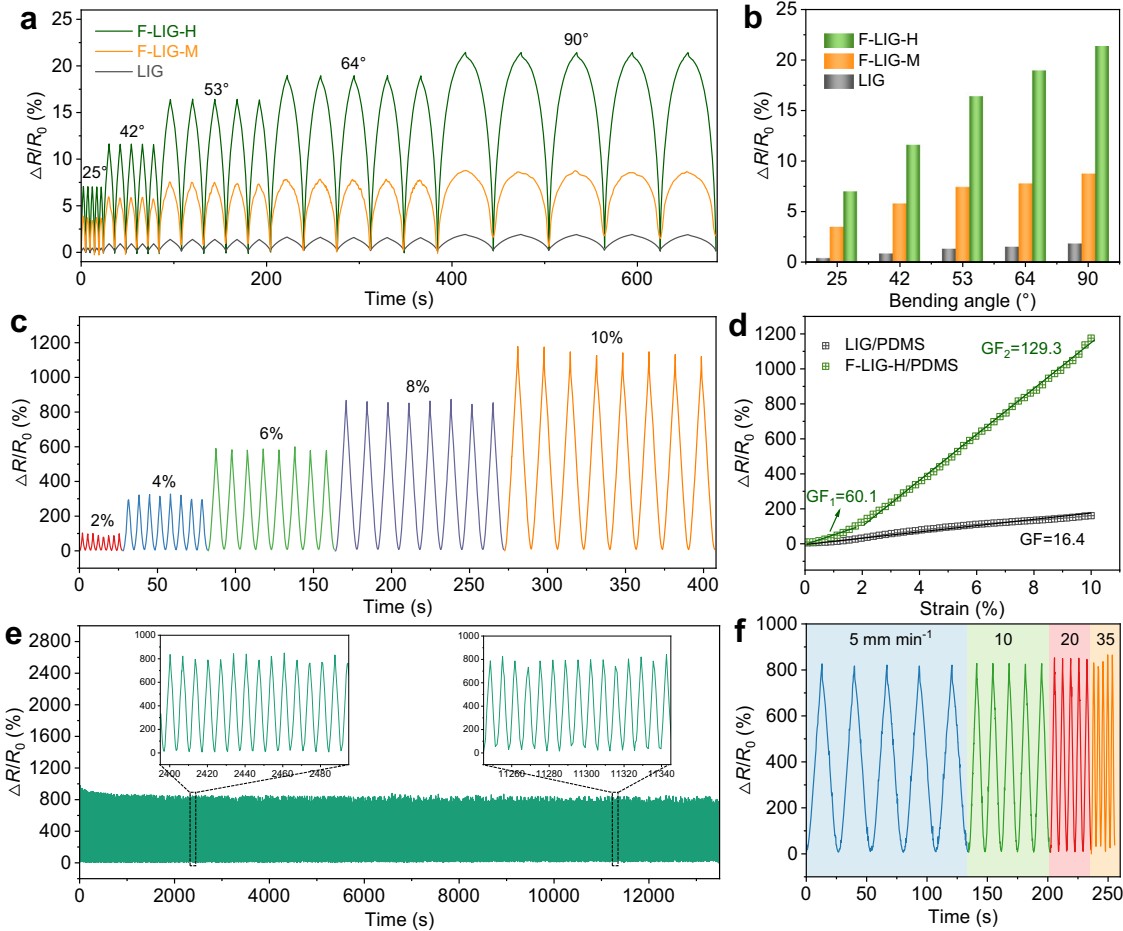

**Fig. 4 | Performance of LIG- and F-LIG-based strain sensors. a** Relative resistance variation of LIG, F-LIG with a high degree of defect healing (F-LIG-H), and F-LIG with a moderate degree of defect healing (F-LIG-M) during the cyclic bending test. **b** $\Delta R/R_0$ value of LIG, F-LIG-H, and F-LIG-M sensors under different bending angles. **c** Relative resistance variation of F-LIG-H/polydimethylsiloxane (PDMS) stretchable sensors at different strain loadings. **d** $\Delta R/R_0$-to-strain relationship of LIG/PDMS and F-LIG-H/PDMS. The gauge factor (GF) was determined for LIG/PDMS as GF, while for F-LIG-H/PDMS it was determined as $GF_1$ and $GF_2$ in different loading regions. **e** Stability test of F-LIG-H/PDMS under 8% strain. Insets are enlarged views of the relative resistance variation during time intervals of 2395 – 2495 s and 11245 to 11345 s. **f** Relative resistance variation of F-LIG-H/PDMS under 8% strain with different stretching rates.

## Application of F-LIG as strain sensors

By virtue of the significantly improved conductivity and well-preserved 3D porous architectures in F-LIG, its application as high-performance piezoresistive strain sensors were demonstrated. For the fabrication of the devices, LIG and F-LIG patterns measuring 1 mm × 20 mm were utilized. Initially, the resistance of the LIG pattern is measured to be ~1300 Ω. To trigger the FJH process in these samples, higher FJH voltages are needed. Specifically, FJH voltages are set at 430 V and 410 V, respectively, resulting in F-LIG with a high degree of defect healing (referred to as F-LIG-H) and a moderate degree of defect healing (referred to as F-LIG-M). The resistance of F-LIG-H and F-LIG-M is determined to be ~270 Ω and ~850 Ω, respectively. Here, the voltage and current curves, as well as the power and energy density, are also recorded and calculated for reference (Supplementary Fig. 18). Due to the limits in the rising speed of our power supply, the actual voltage cannot reach the set value for voltages exceeding 200 V. When the voltage is set at 430 V, the actual voltage maxes out at 380 V. The resulting $P_A$ and $Q_A$ reaches ~1900 W cm$^{-2}$ and 25.72 J cm$^{-2}$, respectively. These values are comparable to those obtained from the 1 mm × 10 mm patterns treated under 190 V. At 410 V (actual voltage ~370 V), the $P_A$ and $Q_A$ reaches ~900 W cm$^{-2}$ and 14.21 J cm$^{-2}$, respectively, which is lower than the optimal case. Consequently, the resistance decrease is moderate under this condition. The as-prepared LIG and F-LIG patterns were then directly used as strain sensors to detect bending

deformation because of the inherent flexibility of PI substrate. Different bending angles (25°, 42°, 53°, 64°, and 90°) were achieved by adjusting the distance between the clamps (Supplementary Fig. 19). The relative resistance variation ($\Delta R/R_0$, $\Delta R$ and $R_0$ stands for the resistance variation and the initial resistance value, respectively) was recorded, where the applied voltage was maintained constantly at 1 V. As shown in Fig. 4a, during the cyclic bending deformation, the samples consistently display distinct and replicable resistance response. As the bending angle increased, the resistance exhibits a continuous and gradual ascent. This behavior can be attributed to the decreased contact between adjacent graphene sheets and thus reduced conductive pathways, caused by the bending leaded in-plane strain. Compared to LIG, F-LIGs exhibit more pronounced change in resistance under the same bending angle. For example, at the bending angle of 90°, F-LIG-H and F-LIG-M reach a $\Delta R/R_0$ value of 21.37% and 8.82%, respectively, which is 11.2-fold and 4.6-fold as large as that of original LIG (1.91%) (Fig. 4b).

For strain sensors, stretchability is a crucial property to meet the requirements of diverse application scenarios. Consequently, we proceeded to transfer the LIG and F-LIG-H patterns onto polydimethylsiloxane (PDMS) substrate to fabricate stretchable strain sensors. The strain sensing performance was evaluated under various strain loads. As depicted in Fig. 4c and Supplementary Fig. 20, similar to the bending sensors, both LIG/PDMS and F-LIG-H/PDMS exhibit an

increasing trend in resistance with applied strain. As the deformation increases, there is a decrease in the contact between graphene sheets, and in some cases, the strain can even cause the graphene cell walls to rupture. These reduced contact and structural damage lead to hindrances in conductivity, resulting in an increase in resistance. Given the same mechanism, there are distinct differences in the resistance variation degree between LIG/PDMS and F-LIG-H/PDMS under the same strain levels. At the strain of 10%, the F-LIG-H/PDMS sensor demonstrates a prominent $\Delta R/R_0$ value exceeding 1180%, whereas the LIG/PDMS only reaches a $\Delta R/R_0$ value of 163%, indicating that the former exhibits significantly higher sensitivity to strain. In Fig. 4d, the resistance variation with applied strain of LIG/PDMS and F-LIG-H/PDMS are plotted to determine the gauge factor (GF), which is defined as the first derivative of $\Delta R/R_0$ with respect to strain. As shown in Fig. 4d, the GF of LIG- and F-LIG-based strain sensors are determined to be 16.4 and 129.3, respectively. The substantial improvement in sensitivity observed in the F-LIG/PDMS strain sensor can be attributed to the improved conductivity and well-preserved 3D porous architectures of graphene scaffolds. These characteristics facilitate more efficient electron transport, thus resulting in more pronounced changes in resistance under deformation. Besides, the enlarged pore size of F-LIG, as depicted in the SEM images, may also contribute to the enhanced sensitivity, because larger pore sizes can amplify the disconnection under strain, enabling more significant resistant changes. Similar finding has been reported in the literature for graphene-based material[49]. The performance of our F-LIG-based strain sensor along with recently reported piezoresistive strain sensors are compared in Supplementary Table 1, with a specific focus on low strain ranges. Our sensor exhibits relatively high GF among state-of-the-art LIG-based sensors and outperforms many metal- and other low-dimensional nanomaterials-based sensors, highlighting its competence in detecting subtle deformations with high accuracy.

To assess the reliability and stability of the strain sensors, first we compare the resistance variation with strain during the stretching and releasing process (Supplementary Fig. 21). It can be observed that these two paths closely align with each other. This result indicates that the sensors exhibit negligible hysteresis, which can greatly minimize the inconsistencies and errors in the measurement. Additionally, tests were conducted at various tensile speeds ranging from 5 – 35 mm min$^{-1}$. As shown in Fig. 4f, the sensing signals exhibit consistent amplitudes that are independent of the deformation frequency. This observation indicates that the strain sensors can maintain their performance and sensitivity across different tensile speeds, ensuring reliable and accurate measurements. Furthermore, the durability of the strain sensors is demonstrated through long-time cyclic tensile test. As depicted in Fig. 4e, the resistance variation across 2000 tensile cycles and 13000 s remains stable. This result supports the sensors' resilience and robustness, demonstrating their suitability for practical applications where they can withstand frequent and extended use without noticeable degrading performance.

The high sensitivity of the stretchable sensors, particularly in the small strain ranges, enables their successful application in monitoring subtle human body motions, recognizing phonation, information encryption and communication, and human-machine interfaces. In Fig. 5a and b, we demonstrate the monitoring of various human motions, including eye blinking, mouth opening and closing. The sensor accurately captures and tracks these movements, showcasing its capability for precise motion detection. Additionally, the high sensitivity allows the sensor to detect even more subtle movements, such as sounds vibrations and wrist pulse. The F-LIG-H/PDMS strain sensor proves effective in detecting sound vibrations when attached to the microphone of a smartphone. Figure 5c presents distinct and highly repeatable resistance variation signals induced by different sound vibrations, including "good", "hello", "graphene", and "sensitivity". Figure 5d illustrates the recorded resistance vibration of the F-

LIG-H/PDMS strain sensor, which precisely matches the subject's wrist pulse. Remarkably, the sensor is capable of distinguishing the pulse signal vibrations associated with the percussion wave (P-wave) and diastolic wave (D-wave), as depicted in Fig. 5e. These results highlight the versatile applications of the F-LIG-H/PDMS strain sensor, demonstrating its great potential in various fields such as healthcare and audio sensing.

In the application of wearable human-machine interfaces, the LIG and F-LIG-H were mounted on the finger positions of a rubber glove, respectively, forming smart gloves. Each sensor was wired to a testing circuit to realize the motion recognition of human fingers and the real-time control of robotic hand. Figure 5f shows the schematic diagram of the testing circuit. The signal acquisition part primarily involves a voltage divider circuit, comprising five pairs of LIG or F-LIG-H sensor ($R_s$) and the corresponding constant resistor ($R_c$). As the sensors are bent and straightened, the resulting resistance variation can be converted into the output voltage signals ($V_{out}$), which are the voltages across the $R_c$ and can be calculated through the equation $V_{out} = V_{source} \times \frac{R_c}{(R_c + R_s)}$. The $V_{out}$ signals are then sent to an Arduino board for the control of the robotic hand. To ensure a fair comparison of sensitivity between the LIG and F-LIG-H gloves, parameters in the control module were kept consistent throughout the experiments. However, due to the differing initial resistances of LIG and F-LIG-H, different $R_c$ were employed to minimize the impact of the sensitivity drifting in the testing circuit. Specifically, constant resistors of 11.5 k$\Omega$ and 5.5 k$\Omega$ were paired with each LIG and F-LIG sensors, respectively. This arrangement ensures that the $V_{out}$ can attain the same value when the LIG or F-LIG sensors are in their initial resistance state, i.e., when the fingers are straightened. Photos illustrating the response of robotic hand controlled by the F-LIG-H and LIG smart gloves are shown in Supplementary Fig. 22. In both tests, the user performs the same hand motion by bending one finger. It can be found that the F-LIG-H glove can accurately control the robotic fingers to mimic the user's finger bending and quickly reach the complete bending status. However, there is very little bending visible when the robotic hand is controlled by LIG sensors (refer to Supplementary Movie 1). This observation visually demonstrates the higher sensitivity of F-LIG-based sensors compared to LIG-based ones. Thus, real-time control of the robotic hand to perform various hand gestures, including "yeah", "six", "fist", and "okay" can be successfully realized by the F-LIG-H smart sensing glove (Fig. 5g-j and Supplementary Movie 2). The quick and accurate response of the robotic hand demonstrates the high feasibility of F-LIG strain sensors in hand gesture recognition and human-machine interaction. Besides, the precise and real-time control offered by the smart glove makes it possible to define the duration and frequency of bending signals to transmit Morse code. As illustrated in Fig. 5k and l, by manipulating the endurance and frequency of finger bending, words such as "SOS" and "HELP" can be demonstrated, enabling information encryption and communication.

## Application of F-LIG in low-voltage sterilization

In addition to strain sensors, we also explored the potential of F-LIG in low-voltage sterilization. In this section, square LIG films with dimensions of 10 mm × 10 mm were utilized. Through the FJH treatment, a notable decrease in resistance is observed from the initial value of ~250 $\Omega$ to 70 $\Omega$ (Supplementary Fig. 11). To quantitatively assess the antibacterial performance, a colony-forming unit (CFU) assay was conducted using *Escherichia coli* (*E. coli*) as model bacterium. Initially, there is a similar adherence and survival of *E. coli* on both LIG and F-LIG, with a count of ~8.5 × 10$^4$ CFU mL$^{-1}$. Upon the application of low DC voltages, significant sterilization is observed (Supplementary Fig. 23 and 24). Notably, F-LIG exhibits a higher bacterial killing rate compared to LIG at each applied voltage. For instance, at 5 V, LIG shows a moderate bactericidal activity of 76.3%, while F-LIG achieves an excellent efficiency of 99.94%. Remarkably, the viable count of *E.*

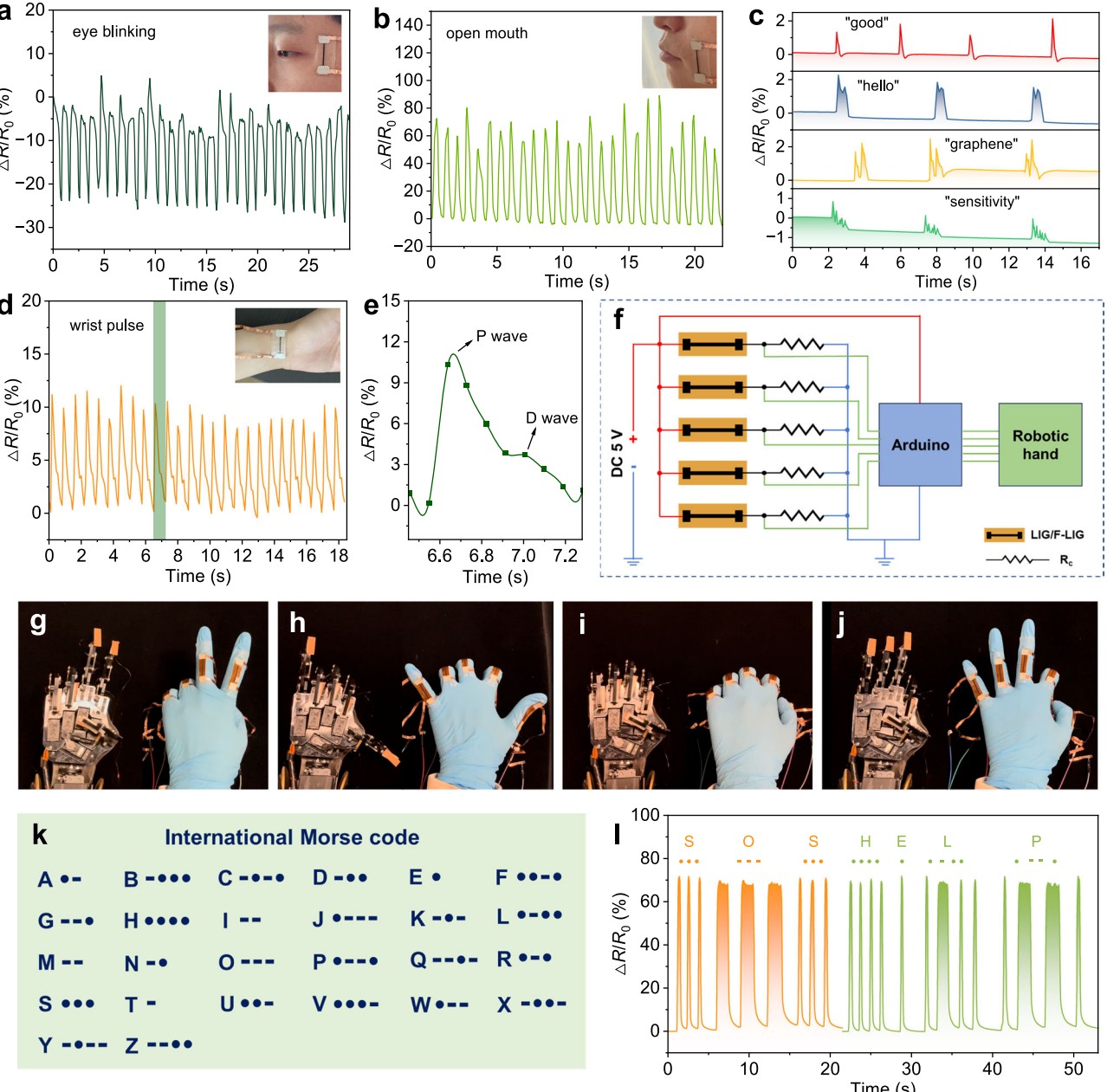

**Fig. 5 | Application demonstration of F-LIG-based strain sensors.** Real-time resistance response of the F-LIG-H/PDMS sensor for the detection of (**a**) eye blinking, (**b**) mouth opening, (**c**) microphone phonation, and (**d**) wrist pulses. Insets are the photographs of the sensor attached to different parts of the subject's body. **e** Enlarged view of the shaded area in (**d**) demonstrating the distinguishable P-wave and D-wave. **f** Schematic diagram of the testing circuit of robotic hand controlling. **g**–**j** Digital photographs illustrating the control of a robotic hand to make various gestures by the F-LIG-H sensors-integrated smart glove. **k** International Morse code. **l** Morse code for "SOS" and "HELP" produced by finger bending.

*coli* remains $2.09 \times 10^4$ CFU mL$^{-1}$ on LIG, whereas significantly reduced to only ~53 CFU mL$^{-1}$ on F-LIG. During the process of electric sterilization, the significantly enhanced conductivity of F-LIG leads to a relatively higher current density compared to LIG under the same applied voltage (Supplementary Fig. 25). Consequently, F-LIG films are capable of achieving higher surface temperatures (Supplementary Fig. 26). However, it is worth noting that the surface temperature remains moderate, with the F-LIG film reaching only 57 °C under 5 V. It has been reported that at moderate temperatures and short durations, electrical current is the primary factor responsible for bacterial killing, with the amount of bacterial eradication dependent on the current density[50]. The mechanism of electrical sterilization is believed to involve the electroporation of bacteria and direct electron transfer between the bacteria and LIG.

In summary, we managed to couple the LIG with the FJH technique to fabricate flexible graphene patterns with reduced defect density and improved conductivity. The improvements in atom arrangement and structure were confirmed by various characterizations, such as Raman spectroscopy, XRD, PDFs, and HRTEM. The enhanced conductivity and well-preserved 3D scaffolds allow for the development of strain sensors based on F-LIG, which exhibited an 8-fold increase in sensitivity (GF ≈ 129) compared to the original LIG-based sensors (GF ≈ 16). Applications in the sensitive detection of subtle human motions, phonation recognition, information encryption and communication, as well as human-machine interfaces are successfully achieved. Furthermore, application of F-LIG in low-voltage sterilization is also demonstrated. In summary, the combination of LIG with FJH offers a straightforward, efficient, and versatile solution to address the

inherent structural defects in LIG. This approach holds significant promise for advancing high-performance graphene-based electronics across various fields and could inspire more promising applications such as supercapacitors and chemical sensors.

## Methods

### Fabrication of LIG and F-LIG

PI film with a thickness of 250 μm (Zeman Tape Material Technology, China) was directly subjected to laser irradiation on a $CO_2$ laser cutting platform (Minsheng Laser #MSDB-FM60 $CO_2$ Laser Marker, 60 W) with a 10.6 μm wavelength in ambient condition to fabricate LIG patterns. The laser was operated in vector mode and the power, frequency, duty cycle ratio, scan speed, pulses/dot, and line space were set as 4.8 W, 10 kHz, 8%, 1000 mm/s, 5 and 30 μm, respectively. For other substrates, PEEK film with thickness of 250 μm was provided by Meideyuan Plastic Products Co., LTD, China. The PES and PES/lignin films were fabricated using solution casting method. In a typical procedure, 1.2 g of PES commercial polymer powders were dissolved in 10 mL dimethylformamide (DMF). The resulting homogenous solution was poured into an aluminum dish (inner diameter 6 cm) and kept at 80 °C overnight to obtain PES films. To prepare PES/lignin film, an additional 0.8 g of lignin was added to the PES solution, and the casting was carried out in the same manner. The FJH process was conducted in a vacuum chamber, by applying DC voltages to the LIG patterns using a programmable switching power supply (ITECH IT6500C). Dumbbell shaped LIG patterns with an effective region of 1 mm × 10 mm and 1 mm × 20 mm were employed for the crystalline structure investigation and strain sensor fabrication, respectively. Temperatures reached during the FJH process were measured using an IR thermometer (Micro-Epsilon), which determined the temperature by fitting the emitted black-body radiation.

### Characterizations

The resistance of LIG patterns before and after FJH were measured by two-point probe method using a UNI-T UT39C+ multimeter. Measurement of several equivalent samples were done to obtain an average. Raman spectroscopy was performed on a WITec RAMAN alpha 300 R system and the excitation wavelength was 532 nm. The microstructure of the LIG and F-LIG samples were examined by a QUATTRO S SEM with an operating voltage of 15 kV. Morphologies at the atom scale were evaluated by HRTEM images were acquired on a Cs-corrected S/TEM JEOL ARM 300F2 with an acceleration voltage of 80 kV. XPS spectra were acquired using a Thermo ESCALAB 250Xi spectrometer. XRD patterns were collected by a powder Rigaku X-Ray diffractometer with Cu Ka radiation (λ = 1.54 Å).

The PDFs were extracted from high energy synchrotron X-ray total scattering by direct Fourier transform of a reduced structure function (F(Q), up to Q ≈ 24.7 Å$^{-1}$) using the 11-ID-C beamline at the Advanced Photon Source (APS) of Argonne National Laboratory (X-ray wavelength 0.1173 Å). For each sample, the powder was loaded into a capillary about 2 mm in diameter. The measurement of each sample was repeated three times, with a total data acquisition time of 9 min. The background scattering from the empty capillary was also extracted during measurements. For data processing, the two-dimensional diffraction images collected were first converted to one-dimensional scattering profiles by GSAS-II. Then, PDFgetX3 was used for the background subtraction and the normalization of the atomic form factors in reducing I(q) to the structure function S(q). Finally, S(q) was Fourier transformed to G(r), a pair distribution function in real space.

### Strain sensor fabrication

The as-prepared LIG and F-LIG (1 mm × 20 mm) with PI substrates were directly utilized as bending sensors by connecting copper wires to both ends of the patterns using silver paste. For the fabrication of stretchable strain sensors, LIG and F-LIG patterns need to be transferred onto an elastic PDMS substrate to overcome the inextensibility nature of PI. The PDMS prepolymer and curing agent (Sylgard 184, Sigma-Aldrich) were thoroughly mixed by a weight ratio of 10:1. After degassing, the mixture was casted onto the surface of PI with LIG or F-LIG patterns. The coated PI film was then cured at 80 °C for 2 h to allow the PDMS to solidify. After curing, the PI film was carefully peeled off, leaving behind the LIG/PDMS or F-LIG/PDMS composites. Then silver paste and copper wires were applied to the connecting pads in the same manner as the bending sensor. The transferred conductive patterns were then enclosed in another layer of PDMS and cured, exposing the copper wires for electrical connection.

### Electromechanical test

The bending and tensile test was conducted using an Instron Series 3382 UTM System. In the case of bending sensing, LIG and F-LIG on PI substrates were adhered to another strip of PI film with a size of 1 cm × 12 cm. The bending region was kept at 1 cm × 8 cm after clamping. The specimen was bent by moving the crosshead toward each other, at a speed of 200 mm min$^{-1}$. When it comes to the tensile test, strain was applied longitudinally, and the crosshead's moving speed was set at 10 mm min$^{-1}$ throughout the test unless otherwise stated. During the dynamic bending or stretching, real-time measurements of the sensors' resistance signals were made using a Keithley 2612B source meter.

### Human motion detection, phonation recognizing and robotic hand control

To detect human motions and sound vibrations, the prepared F-LIG-H/PDMS strain sensors were attached to different parts of a subject's body, such as the wrist and face, or positioned on the microphone of a cell phone. The resistance signals generated during the movements were captured by a Keithley 2612B source meter. For the real-time control of robotic hand, five LIG or F-LIG-H bending sensors were mounted to the knuckle positions of a rubber glove. The sensors were then wired to a control circuit, which processes the signals from the sensors and sends the control signals to the robotic hand, so that it can mimic and replicate the movements of the wearer's hand in real-time.

### Low-voltage sterilization experiments

A single colony of *E. coli* on a solid nutrient agar (Oxoid, CM0003) plate was transferred to 5 mL of nutrient broth (Oxoid, CM0001) medium in a shaking incubator (170 rpm) at 37 °C overnight. The LIG and F-LIG films with dimensions of 10 mm × 10 mm were randomly divided into control groups and experimental groups and each group had at least three films. First, the films were immersed into the *E. coli* suspension ($10^8$ CFU mL$^{-1}$) and incubated at 37 °C for 1 h. Next, the films were removed from the suspension and washed with ringer solution to remove unattached bacteria. Then, variable DC voltages (1 V, 3 V, and 5 V) were applied to the films for 2 min by a Keithley 2612B source meter. After that, the films were sonicated in 10 mL ringer solution for 5 min (10% power) to detach the bacteria. Bacteria CFU were enumerated using the plate counting method.

### Reporting summary

Further information on research design is available in the Nature Portfolio Reporting Summary linked to this article.

## Data availability

The data that support the findings of this study are available from the corresponding author upon request. Source data are provided with this paper and can be accessed at https://doi.org/10.6084/m9.figshare.25466926. Source data are provided with this paper.

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

## Acknowledgements

The study described in this paper was partially supported by the Shenzhen Science and Technology Program (JCYJ20220818101204009), Fisheries Enhancement Fund (FEF22004), CityU Applied Research Grant (No. 9667254), State Key Laboratory of Marine Pollution Seed Collaborative Research Fund (SKLMP/SCRF/0060) and Hong Kong Research Grants Council (11309723). B.Z.T. acknowledges support from the Shenzhen Key Laboratory of Functional Aggregate Materials (ZDSYS20211021111400001), the Science Technology Innovation Commission of Shenzhen Municipality (KQTD20210811090142053, JCYJ20220818103007014), and the Innovation and Technology Commission (ITC-CNERC14SC01). X.Y. thanks the support from Research Grants Council of the Hong Kong Special Administrative Region (Grant Nos. 11213721, 11215722) and City Universiy of Hong Kong (Grant Nos. 9678274). This research used resources of the Advanced Photon Source, a U.S. Department of Energy (DOE) Office of Science User Facility, operated for the DOE Office of Science by Argonne National Laboratory under Contract No. DE-AC02-06CH11357.

## Author contributions

R.Y. conceived the idea and designed the experiments. R.Y., B.T., Y.R., F.C., and X.Y. supervised the research. L.C. conducted the majority of the experiments. C.Yeung, L.H., G.Y. and H.S. contributed to some of the experiments. Y.R. and J.Y. performed and analyzed the X-ray scattering experiment. W.L. captured the HRTEM images. C.Yiu performed the robotic hand control experiment. R.Y. and L.C. analyzed the data and wrote the manuscript with input and comments from the other authors.

## Competing interests

R.Y. and L.C. have filed a Chinese Patent Application through City University of Hong Kong Shenzhen Research Institute. The patent application encompasses the fabrication methods and applications of F-LIG in this manuscript. The remaining authors declare no competing interests.
