## [Peer Review File · Nature Communications]

Flash Healing of Laser-induced GrapheneREVIEWER COMMENTS

Reviewer #1 (Remarks to the Author):

This manuscript presents a very interesting approach based on Flash Joule Heating for enhancing the properties of Laser Induced Graphene (LIG) obtained from polyimide (PI). Through the application of a short (20 ms) DC voltage, a Joule heating of LIG patterns is produced which enables a notable increase of LIG conductivity. This is ascribed to an enrichment of order and graphene crystallite size and an overall reduction of defects in LIG. The paper investigates in depth the cause of this enhancement and its dependence on applied voltage during the FJH treatment, using Raman spectroscopy, HRTEM, XPS, SEM, providing convincing evidences.

The resulting material (F-LIG) is then used for creating strain sensors (bending, supported on PI, and stretchable, transferred on PDMS) with gauge factor and sensitivity surpassing the state of art.

Overall this is a very interesting contribution with relevant findings which are interesting both from a basic point of view and for the applications (sensors). It is also appealing to a broad readership.

There are some **major questions and concerns which authors could probably address** and that I suggest to clarify before to resubmit the manuscript and before being able to accept the manuscript for publication.

1) The effect of applied voltage(current) is poorly investigated and described. Indeed, while voltage is the parameter changed, the FJH is based on heating, so I would have expected a more in depth investigation in Temperature. No idea is given about the temperature reached during the application of current, which is responsible (probably?) for the rearrangement of LIG and the enhancement of its properties. No thermal camera imaging or other means of providing an idea of temperature (modeling? Spectral analysis?) is provided. I am wondering why.

2) at page 5 (line 107) the very qualitative term “ultrahigh temperature” is used. I would avoid it, as much as possible and instead place some estimate (if not a direct measurement)

3) Also, in part connected to Q1, while voltage is applied, the heating is due to the resulting current. Probably a notable parameter in FJH is the electrical power. Indeed the generated Joule heating is directly related to this (dissipated) electrical power which is transformed into heat. Probably rediscussing everything based on power can give more insight?

4) There is not much description on the factors which led to the selection of 20 ms for the duration of FJH. Why? were other durations studied? Why not shorter? Or Longer? Or...repeated pulses? Without these descriptions the whole thing is not really described and investigated. I would suggest to discuss the effect of pulse duration (and also show results of experiments at various durations?)

5) Similar to Q4: why those voltages were selected? Why not larger? Or smaller? Again, here, probably the concept of electrical power can be useful for clarifying.

6) page 10, line 191: “...rapid heating and cooling during laser irradiation in the nanosecond/picosecond timescale”. Are you sure about this? The laser used has a pulse frequency of 10KHz. A better insight in the timescale of processes would be needed. Also, compared to the FJH and to the abundant literature

on LIG produced with different pulsed laser sources.

7) There are many typos (e.g. “angel” instead of “angle”, and “connect” instead of “context” and so on) which I recommend to check and change. Also the meaning of some sentences sounds a bit obscure or ambiguous, probably for a language/style reason (e.g. line 126 “under 180V” what does it mean?, line 129 “second half”, second half of what? No mention of first half...and so on).

Reviewer #2 (Remarks to the Author):

The author proposed a method to process graphene through flash healing, and the resulting graphene has lower initial resistance and is thus more suitable as a strain sensor. However, compared to LIG, the innovation of this work is not significant. Essentially, it also utilizes high temperatures to increase the carbon content in the material. Moreover, the performance does not show any advantage over traditional strain sensors. Additionally, regarding the application, the function realized in this work has been demonstrated in many articles, indicating weak innovation. Furthermore, unless an extremely excellent performance is achieved, a single-functional strain sensor cannot meet the requirements of high-quality papers by Nature Communications, compared to electronic devices integrating machine learning algorithms or multi-functionalities. Therefore I don't recommend the acceptance of this paper. Some specific suggestions are as follows:

1.The introduction section requires further refinement. Currently, it lacks clarity in its logical flow, and it fails to distinctly highlight the research problem, objectives, and strategies. The introduction should explicitly present the research problem while elaborating on its significance, specific implications, and application areas.

2.A thorough grammatical review is warranted, encompassing verb agreement, spelling accuracy, proper usage of modal verbs with infinitives, and maintaining a refined and consistent language expression. Ensuring uniformity in abbreviations, particularly in figure captions, is crucial. Additionally, thorough spell-checking is necessary, such as correcting "bending angel" to "bending angle."

3.The reasons behind the structural integrity of F-LIG samples with fewer defects and the underlying mechanisms need clarification. It is important to differentiate your study from those referenced in Nano Lett. 2016, 16, 11, 7282–7289 and ACS Nano 2023, 17, 3, 2506–2516 and emphasize the uniqueness of your research in this regard.

4.A legend is missing in Figure 1 (d), and it should be included for a clear understanding of the figure's content.

5.Explanations are required for the variation in applied voltages, with initial XPS and SEM analyses at 150-190 V during FJH treatment, and subsequent applications at 410 V. Similarly, the adjustment of XRD characterization to use a direct current source at 130 V for analysis needs clarification to ensure the

rationale behind these choices is explained.

6.The application section of the article lacks innovation. (Liquid metal-based strain-sensing glove for human-machine interaction, 2023).

Reviewer #3 (Remarks to the Author):

the manuscript of Cheng et al describes the formation of low resistance LIG structures by a simple flash joule heating process applied to fabricated LIG tracks and powders. This process sensibly reduces the resistance of LIG by 5 times thus opening sensitive applications in health and electronics. The findings are very impressive and the manuscript well written, with relevant data provided to support findings and claims. the methodology is sound and the results are highly novel and interesting for the field.

I would recomend publication of the manuscript after addressing the following points:

it would be interesting to know if the flash healing step works with substrates that are different than polyimide or if it would be too harsh. although not mandatory, this extra information would open up applications in green electronics.

although the performance of strain sensor and other sensors shown in Fig 5 are greatly improved compared to LIG , it would be good to add a comparative table (may even in the SI).

Response to Reviewers

To make it clear, we have used *italic* for the reviewers' comments, **bold** for our replies and blue for our revisions in the manuscript.

Response to Reviewers' comment

Reviewer #1

Comments:

This manuscript presents a very interesting approach based on Flash Joule Heating for enhancing the properties of Laser Induced Graphene (LIG) obtained from polyimide (PI). Through the application of a short (20 ms) DC voltage, a Joule heating of LIG patterns is produced which enables a notable increase of LIG conductivity. This is ascribed to an enrichment of order and graphene crystallite size and an overall reduction of defects in LIG. The paper investigates in depth the cause of this enhancement and its dependence on applied voltage during the FJH treatment, using Raman spectroscopy, HRTEM, XPS, SEM, providing convincing evidences.

The resulting material (F-LIG) is then used for creating strain sensors (bending, supported on PI, and stretchable, transferred on PDMS) with gauge factor and sensitivity surpassing the state of art.

Overall this is a very interesting contribution with relevant findings which are interesting both from a basic point of view and for the applications (sensors). It is also appealing to a broad readership.

There are some major questions and concerns which authors could probably address and that I suggest to clarify before to resubmit the manuscript and before being able to accept the manuscript for publication.

Reply: We thank the reviewer for the positive comments on the scientific merits and constructive suggestions for further improvement. Please see below our point-by-point responses to the comments.

1. The effect of applied voltage(current) is poorly investigated and described. Indeed, while voltage is the parameter changed, the FJH is based on heating, so I would have expected a more in depth investigation in Temperature. No idea is given about the temperature reached during the application of current, which is responsible (probably?) for the rearrangement of LIG and the enhancement of its properties. No thermal camera imaging or other means of providing an idea of temperature (modeling? Spectral analysis?) is provided. I am wondering why.

Reply: Thank you for the suggestion to improve the paper. We agree well with the reviewer that the rearrangement of carbon atoms and the flash healing effect depend on the temperature reached during the flash Joule heating (FJH) process, while the voltage acts as the control parameter for tuning the temperature. However, due to the intense and transient nature of the heating, it is challenging for infrared (IR) cameras to capture the temperature accurately. Following your suggestion and the approaches described in previous literatures on FJH, we conducted the temperature measurement using an IR thermometer (Micro-Epsilon), which determined the temperature by fitting the emitted black-body radiation. As shown in Figure R1, the temperatures achieved under the applied voltages (150-190 V) were measured to be around 1300, 1700, 2100, 2300, and 2500 °C, respectively. By integrating the measurements of temperature, resistance, and Raman analysis, a reasonable conclusion can be reached: the healing of defects initiates at around 1700 °C. With the further temperature escalation beyond this critical point, notable decrease in resistance and defect density becomes evident.

Figure R1. Temperature reached during the FJH process at different voltages.

In the revised manuscript, we have added the above results. Figure R1 has been included as part of the revised Figure 1c.

2. At page 5 (line 107) the very qualitative term “ultrahigh temperature” is used. I would avoid it, as much as possible and instead place some estimate (if not a direct measurement).

Reply: Thank you for pointing out the lack of precision in our description. As described in the response to Q1, measurements of the temperature reached during the FJH process were conducted. Accordingly, we have made revisions to replace any qualitative expressions with precise and quantitative descriptions throughout the manuscript.

3. Also, in part connected to Q1, while voltage is applied, the heating is due to the resulting current. Probably a notable parameter in FJH is the electrical power. Indeed the generated Joule heating is directly related to this (dissipated) electrical power which is transformed into heat. Probably rediscussing everything based on power can give more insight?

Reply: Thank you for the excellent suggestion to improve the paper. As mentioned in the response to Q1, the temperature achieved during the FJH process is indeed the primary factor influencing the effectiveness of defect healing. In this study, we employed different FJH voltages to regulate the temperature, which is essentially a result of both voltage and current, specifically the electrical power consumed within the LIG patterns. Consequently, we recorded the instantaneous voltage (U) and instantaneous current (I) during the FJH process. These measurements enable precise calculations of power and energy, providing further insights into the relationship between the applied voltage and the FJH process. Considering that LIG patterns with different dimensions were used in our study, we derived the instantaneous areal power density (P_A) and areal energy density (E_A , also the generated Joule heat density) using the equations $P_A = UI/A$ and $E_A = UI/Adt$, respectively. Here A represents the area of the LIG patterns, and t denotes the discharging time.

Figure R2 illustrates the recorded voltage and current curves, as well as the calculated areal power density and energy density, for LIG patterns with dimensions of 1 mm × 10 mm. The voltage is programmed with a switch-on duration of 20 ms. Under each applied voltage, the resulting current and power density exhibit similar profiles. As the voltage increases, there is an upward trend observed in the current, power density, and energy density. Notably, when the FJH voltage is 190 V, the P_A and corresponding E_A can reach ~2100 W cm⁻² and 27.55 J cm⁻², respectively, indicating rapid and significant energy input and heat generation within the LIG patterns. As mentioned in the original manuscript, under 150 V and 160 V (corresponding to an E_A value of 10.5 J cm⁻² and 13.4 J cm⁻², respectively), the current and power density are proportional and exhibit similar profile to the voltage. As voltages increased to 170 V (E_A ~17.6 J cm⁻²) or higher, the current and power density curves initially follow a similar trend as the voltage but then experience a sudden surge, indicating a notable reduction in the resistance of the LIG patterns. Additionally, it is observed that as the increase of FJH voltage, the onset of current surge happens earlier, and the degree of surge also increases. These observations suggest the presence of a critical energy density required to trigger the reorganization of carbon atoms.

Higher voltage can enhance the flash healing effect by generating more heat and reaching higher temperatures.

Figure R2. (a) Voltage, (b) current, and (c) areal power density profiles during the FJH process with different voltages. (d) Areal energy density of the FJH process under different voltages.

For the fabrication of strain sensors, LIG patterns with dimensions of 1 mm × 20 mm were utilized. Initially, the resistance of the LIG pattern is measured to be approximately 1300 Ω. To trigger the FJH process in these samples, the FJH voltages are set at 430 V and 410 V, respectively, resulting in F-LIG with a high and moderate decrease in resistance. However, it should be noted that, due to the limitations in the rising speed of our power supply, the actual voltage cannot reach the set value for voltages exceeding 200 V. As shown in Figure R3, when the voltage is set at 430 V, the actual voltage reached a maximum of 380 V. Consequently, the resulting P_A and Q_A reached approximately 1900 W cm⁻² and 25.72 J cm⁻², respectively. These values are comparable to those obtained from the 1 mm × 10 mm patterns treated under 190 V, suggesting a similar level of FJH process. At 410 V (actual voltage ~370 V), the P_A and Q_A reaches ~900 W cm⁻² and 14.21 J cm⁻², respectively, which is lower than the optimal case. Accordingly, the resistance decrease is moderate under this condition.

Figure R3. Curves of (a) voltage, current, and (b) areal power density and integrated energy density for LIG patterns with dimensions of 1 mm × 20 mm.

Thank you again for the constructive suggestion to enable a deeper understanding of the power dynamics and heat energy generation throughout the FJH process. Figure R2 and R3 have been added to the revised manuscript and Supporting Information, respectively. Related discussions were also added.

4. There is not much description on the factors which led to the selection of 20 ms for the duration of FJH. Why? were other durations studied? Why not shorter? Or Longer? Or...repeated pulses? Without these descriptions the whole thing is not really described and investigated. I would suggest to discuss the effect of pulse duration (and also show results of experiments at various durations?)

Reply: We thank the reviewer's suggestion. As described in the response to Q3, in this work, different voltages were employed to regulate the FJH process through tuning the generated Joule heat. According to the equation $E_A = UI/Adt$, energy density is also positively correlated with the pulse duration. Thus, follow your suggestion, we conducted additional investigations to examine the impact of pulse durations on the FJH process. We applied a constant voltage of 130 V for various durations (20, 30, 40, 50 and 60 ms) to the LIG patterns (1 mm × 10 mm). It should be noted that the lowest temporal resolution achievable by our power supply is 20 ms, which is why we chose it as the shortest pulse duration for our study.

Figure R4 illustrates the curves of instantaneous voltage, current, and power density, as well as the calculated energy density, while Figure R5 shows the resistance and Raman spectra before and after the FJH treatment. When the pulse duration is 20 ms and 30 ms, the current exhibits similar profiles with voltage. Correspondingly, there is minimal change in resistance after the treatment. As the

duration prolongs, an abrupt increase in current is observed during the application of voltage, indicating a decrease in resistance caused by the high temperature-induced defect healing. As illustrated in Figure R5, the resistance shows a slight decrease from $600\ \Omega$ to $530\ \Omega$ at a pulse duration of 40 ms, and further decreased to $150\ \Omega$ when the pulse duration increases to 50 ms. Besides, the D band intensity in the Raman spectra also starts to decrease when the pulse duration reaches 40 ms and becomes significantly lower after the 50 ms FJH treatment. However, further extending heating time to 60 ms leads to the break of the LIG layer, as reflected by the sudden decrease in current.

Figure R4. (a) Voltage, (b) current, and (c) areal power density profiles during the FJH process with different pulse durations. (d) Areal energy density of the FJH process with different pulse durations.

Figure R5. (a) Resistance and (b) Raman spectra of LIG patterns (1 mm × 10 mm) treated under 130 V with different pulse durations.

Indeed, both voltage and pulse duration play crucial roles in regulating the FJH process by controlling the power and energy density. Higher voltage and longer duration result in more Joule heating and more pronounced flash healing effect. According to the equation $E_A = UI/Adt$, with the same voltage increments, shorter pulse durations result in smaller increments in energy density, allowing for finer control over the FJH process. Additionally, shorter heating time also minimizes the heat dissipation. Based on these considerations, the lower limit of the pulse duration (20 ms) was selected to investigate the FJH effect under different voltages.

As for repeated pulses, it is usually employed to maintain an average temperature for an extended period, which has been reported by Hu's Group for continuous thermochemical synthesis (*Nature* 2022, 605, 470–476). Our objective is to achieve a rapid annealing process, thus we opted for a single ultrashort pulse in this study.

In the revised manuscript, Figure R4 and R5 have been added to the Supplementary Information. Related discussions were also added.

5. Similar to Q4: why those voltages were selected? Why not larger? Or smaller? Again, here, probably the concept of electrical power can be useful for clarifying.

Reply: As demonstrated in the response to Q3, there exists a threshold for the energy density or generated Joule heat to trigger the flash healing effect. Below this threshold, corresponding to the cases of 150 and 160 V, the LIG patterns can only be heated to moderate temperatures, which is insufficient to induce the desired atom rearrangement. On the other hand, excessively high voltage, for example, 200 V, leads to excessive transient heating and gas release, resulting in the breakdown of the LIG patterns (Figure R6). From these observations, it can be concluded that proper voltages are needed to meet the threshold energy density for the flash healing process, while also ensuring that the integrity of the LIG layer is maintained.

Figure R6. The broken LIG patterns after the FJH treatment under 200 V 20 ms.

In the revised manuscript, we have included the concept of power and energy density to provide further insight regarding the FJH parameters. Thanks again for your suggestion.

6. page 10, line 191: “...rapid heating and cooling during laser irradiation in the nanosecond/picosecond timescale”. Are you sure about this? The laser used has a pulse frequency of 10KHz. A better insight in the timescale of processes would be needed. Also, compared to the FJH and to the abundant literature on LIG produced with different pulsed laser sources.

Reply: Thanks for your helpful comments. We are sorry for the inaccuracies in our previous description. For a laser operating in pulsed mode, the pulse duration (τ) is determined by the combination of laser frequency (ν) and duty cycle ratio (p) through the equation $\tau = \frac{1}{\nu} \times p$. In our LIG preparation, the frequency and duty cycle ratio were set as 10 kHz and 8%, respectively. Consequently, the resulting pulse duration is 8 μ s. In the revised manuscript, the sentence has been rewritten as “This disorder is due to the amorphous nature induced by the rapid heating and cooling during laser irradiation in the microsecond timescale”.

Additionally, we conducted an extensive literature survey to provide a more comprehensive understanding of the timescales involved in the LIG and FJH processes. As listed in Table R1, to date, a wide range of laser sources with varying pulse durations have been employed for LIG fabrication, mainly including microsecond lasers, nanosecond lasers, picosecond lasers, and femtosecond lasers. The most widely used CO₂ laser is typically characterized by a long pulse duration in the microsecond range. This lengthy pulse duration enables the rapid deposition of a substantial amount of photo-thermal energy in materials to induce LIG conversion.

Table R1. Pules duration of laser sources used in LIG fabrication.

Laser	Wavelength	Pulse duration	Reference
CO ₂ laser	10.6 μ m	14 μ s	Nat. Commun. 2014, 5, 5714
CO ₂ laser	10.6 μ m	14 μ s	Adv. Mater. 2017, 29, 1702211
CO ₂ laser	9.3 μ m	100 ns	Ind. Eng. Chem. Res. 2021, 60, 11161
Yb:YVO ₄ laser	1064 nm	4 ns	Adv. Mater. Technol. 2020, 5, 2000014
Nd:YVO ₄ laser	355 nm	10 ps	Small Methods 2019, 3, 1900208

Yb-doped fiber laser	343 nm	220 fs	Adv. Funct. Mater. 2019, 29, 1902771
Yb-doped fiber laser	1040 nm	255 fs	Adv. Funct. Mater. 2022, 32, 2107768

For the FJH technique, the pulse duration spans a range from tens of milliseconds to seconds, depending on the specific process and materials involved. For instance, a 50 ms pulse duration was utilized in the FJH process to convert pretreated carbon black (CB) and charred coffee grounds into highly crystalline turbostratic graphene (*Nature* 2020, 577, 647-651). In the case of other carbon sources such as anthracitic coal, plastic waste mixtures, and calcine coke, pulse durations of 500-10000 ms were employed to facilitate the formation of high-quality graphene sheets (*Nature* 2020, 577, 647-651; *ACS Nano* 2020, 14, 15595–15604).

In general, laser processing involves pulse durations in the microsecond range or even shorter, while the FJH technique typically operates within the millisecond timeframe. This stark contrast in pulse duration indicates that the electrical pulses used in FJH are at least three orders of magnitude longer than the laser pulses employed in LIG fabrication. Despite the repetition of laser pulses during the LIG fabrication process, the heating duration is still significantly shorter in comparison to the electrothermal annealing process of FJH.

Thanks again for your suggestion. We have incorporated the above time scale comparison into the revised manuscript.

7. There are many typos (e.g. “angel” instead of “angle”, and “connect” instead” of “context” and so on) which I recommend to check and change. Also the meaning of some sentences sounds a bit obscure or ambiguous, probably for a language/style reason (e.g. line 126 “under 180V” what does it mean?, line 129 “second half”, second half of what? No mention of first half...and so on).

Reply: We are sorry for our carelessness and lack of clarity in our previous expressions. In the revised manuscript, we have rectified any typos and made improvements to enhance clarity. Specifically, based on your suggestions in Q3, we have rewritten the discussion part regarding voltage, current, power, and energy during the FJH process to ensure better understanding and readability. The revised section is as follow: “As shown in Fig. 1e-g, the voltage is programmed with a switch-on duration of 20 ms. Under each applied voltage, the resulting current and power density exhibit similar profiles. As the applied voltage increases, the integrated E_A , i.e., the generated Joule heat, as well as the temperature reached, show an increasing trend (Fig. 1c). Specifically, under 150 and 160 V (corresponding to an E_A value of 10.5 and 13.4

J cm⁻², respectively), the current and power density are proportional and exhibit similar profile to the voltage. As voltages increased to 170 V ($E_A \sim 17.6$ J cm⁻²) or higher, the current and power density curves initially follow a similar trend as the voltage but then experience a sudden surge, indicating a notable reduction in the resistance of the LIG patterns. When the FJH voltage is 190 V, there appears a maximum improvement in electrical conductivity. In this case, P_A and corresponding E_A reached ~ 2100 W cm⁻² and 27.55 J cm⁻², respectively, indicating rapid and significant energy input and heat generation within the LIG patterns”.

Reviewer #2

Comments:

The author proposed a method to process graphene through flash heating, and the resulting graphene has lower initial resistance and is thus more suitable as a strain sensor. However, compared to LIG, the innovation of this work is not significant. Essentially, it also utilizes high temperatures to increase the carbon content in the material. Moreover, the performance does not show any advantage over traditional strain sensors. Additionally, regarding the application, the function realized in this work has been demonstrated in many articles, indicating weak innovation. Furthermore, unless an extremely excellent performance is achieved, a single-functional strain sensor cannot meet the requirements of high-quality papers by Nature Communications, compared to electronic devices integrating machine learning algorithms or multi-functionalities. Therefore I don't recommend the acceptance of this paper. Some specific suggestions are as follows:

Reply: We thank the reviewer's comments and suggestions for further improvement of the manuscript. However, we respectively disagree with the comment on novelty. Our group has worked with LIG in the past decade. It has been a challenge in controlling the atomic structures of LIG since its discovery in 2014. Scientifically, we provide insights into how flash heating rectifies the topology of LIG, supported by the neutron scattering, high-resolution images, and other characterizations. As we introduced in the manuscript, the LIG contains polygons, which will be important for applications such as electrocatalysis, where defects are helpful. However, for electronics, in which the LIG is heavily applied, the defects will lead to poorer performance. We use strain sensor as good demonstration of the healing effect, showing significant improvement in strain sensing. In contrast, integrating multiple sensors is not relevant to the main finding of this work. To further show the advantage of healing, we

provide other application in the manuscript, as well as a comparative discussion to other LIG development. We hope the reviewer is satisfied with our revision. Please see below our replies in details.

1. The introduction section requires further refinement. Currently, it lacks clarity in its logical flow, and it fails to distinctly highlight the research problem, objectives, and strategies. The introduction should explicitly present the research problem while elaborating on its significance, specific implications, and application areas.

Reply: Thank you for the comment. In the revised manuscript, the Introduction part has been improved as suggested, adding additional background on defect healing. Our introduction is now structured as follow: (1) traditional methods for graphene synthesis; (2) the introduction of LIG development; (3) the limitation of LIG due to topological defects, and the motivation for defects healing; (4) the existing methods and challenges for defect healing; (5) the introduction of FJH and its potential for rectifying topological defects LIG.

2. A thorough grammatical review is warranted, encompassing verb agreement, spelling accuracy, proper usage of modal verbs with infinitives, and maintaining a refined and consistent language expression. Ensuring uniformity in abbreviations, particularly in figure captions, is crucial. Additionally, thorough spell-checking is necessary, such as correcting "bending angel" to "bending angle".

Reply: Thank you for your correction. We apologize for our careless and inconsistent expressions. Careful proof-reading has been conducted for the revised manuscript to improve the language and ensure consistency throughout.

3. The reasons behind the structural integrity of F-LIG samples with fewer defects and the underlying mechanisms need clarification. It is important to differentiate your study from those referenced in Nano Lett. 2016, 16, 11, 7282–7289 and ACS Nano 2023, 17, 3, 2506–2516 and emphasize the uniqueness of your research in this regard.

Reply: We thank the reviewer's suggestion. LIG is highly appealing for graphene-based electronics for its streamlined fabrication and direct integration into devices. However, it usually suffers from inferior conductivity compared to pristine graphene due to the laser-induced amorphous topologies. These imperfections hinder electron mobility and pose a potential drawback to the performance of electronic devices. So far, the effective and rapid healing of defects in specific LIG patterns remains a challenge.

Flash Joule heating (FJH) is a technique that involves the passage of high direct current (DC) pulses through conductive materials, which enables rapid and intense resistive heating and has found recent applications in the fabrication and processing of carbon materials. For example, Hu's group used FJH to covalently weld carbon nanofibers and enhance graphitization, obtaining highly conductive carbon network (*Nano Lett.* 2016, 16, 11, 7282–7289). Tour's group employed FJH to achieve gram-scale synthesis of high-quality graphene from amorphous carbon black (*Nature* 2020, 577, 647–651). Besides, by controlling the FJH process of carbon nanotubes, hybrids with tunable carbon nanotube/graphene ratio could be obtained (*ACS Nano* 2023, 17, 3, 2506–2516). However, these reports primarily focus on the production of carbon allotropes and lack control over the patterning of the materials.

Inspired by these pioneering works and considering the advantages of rapid, internal, and localized heating over traditional furnace annealing, we believe that FJH holds promise for rectifying topological defects in LIG patterns. In this work, by applying a high-power DC pulse for a microsecond timescale, we managed to achieve the rapid heating of the LIG patterns to an ultrahigh temperature of ~2500 °C. At such extreme temperatures, the amorphous carbon atoms could rearrange to form more regular structures with high degree of graphitization. Advanced characterization techniques, such as Raman spectroscopy, neutron scattering, and atomic-resolution imaging provide compelling evidence of reduced defects and enlarged crystalline domains. The healing of defects facilitates efficient charge transport, leading to a significant enhancement in the conductivity. Additionally, the extremely short duration of the Joule heating within microseconds ensures that the overall shape and internal porous structure of LIG are well preserved. Therefore, this approach holds significant promise for advancing high-performance graphene-based electronics across various fields.

In the revised manuscript, the pioneering works (*Nano Lett.* 2016, 16, 11, 7282–7289, *ACS Nano* 2023, 17, 3, 2506–2516) have been cited and the corresponding sentences have been included in the Introduction section to highlight the uniqueness of our work from previous reported studies. Besides, the Discussion section has been improved to provide a clearer explanation of the flash healing mechanism.

4. A legend is missing in Figure 1 (d), and it should be included for a clear understanding of the figure's content.

Reply: We thank the reviewer's kind reminder. According to you and another reviewer's suggestion, we have replaced the original Figure 1d with Figure R7, as shown below, illustrating the curves of voltage, current, and areal power density (P_A), as well as energy density (E_A) during the FJH process. Furthermore, the related discussions have been rewritten as follow: "To provide further insights into the relationship between the applied voltage and the FJH process, instantaneous voltage (U) and current (I) were recorded during the FJH procedure. Based on these measurements, we derived the instantaneous areal power density (P_A) and areal energy density (E_A) using the equations $P_A = UI/A$ and $E_A = UI/Adt$, respectively. Here A represents the area of the LIG patterns, and t denotes the discharging time. As shown in Fig. 1e-g, the voltage is programmed with a switch-on duration of 20 ms. Under each applied voltage, the resulting current and power density exhibit similar profiles. As the applied voltage increases, the integrated E_A , i.e., the generated Joule heat, as well as the temperature reached, show an increasing trend (Fig. 1c). Specifically, under 150 and 160 V (corresponding to an E_A value of 10.5 and 13.4 J cm⁻², respectively), the current and power density are proportional and exhibit similar profile to the voltage. As voltages increased to 170 V (E_A ~17.6 J cm⁻²) or higher, the current and power density curves initially follow a similar trend as the voltage but then experience a sudden surge, indicating a notable reduction in the resistance of the LIG patterns. When the FJH voltage is 190 V, there appears a maximum improvement in electrical conductivity. In this case, P_A and corresponding E_A reached ~2100 W cm⁻² and 27.55 J cm⁻², respectively, indicating rapid and significant energy input and heat generation within the LIG patterns".

Figure R7. (a) Voltage, (b) current, and (c) areal power density profiles during the FJH process with different voltages. (d) Areal energy density of the FJH process under different voltages.

5. Explanations are required for the variation in applied voltages, with initial XPS and SEM analyses at 150-190 V during FJH treatment, and subsequent applications at 410 V. Similarly, the adjustment of XRD characterization to use a direct current source at 130 V for analysis needs clarification to ensure the rationale behind these choices is explained.

Reply: We thank the reviewer's suggestion. We apologize that we didn't clarify that different voltages were employed due to the variations in size or form of LIG samples.

First, to investigate the effect of FJH process and the structural revolution in LIG, LIG patterns with dimensions of 1 mm \times 10 mm and initial resistance of $\sim 590\ \Omega$ were utilized. As described in the original manuscript, the highest increase in current and the most significant decrease in resistance are observed when the FJH voltage was set to 190 V. As shown in Figure R7, under this condition, the corresponding P_A and E_A reaches a maximum value of $\sim 2100\ W\ cm^{-2}$ and $27.55\ J\ cm^{-2}$, respectively. In the application section, strain sensors were fabricated using longer LIG patterns measuring 1 mm \times 20 mm, which possess an initial resistance of $\sim 1300\ \Omega$. It is important to note that the consumed electrical energy density is the key factor in determining the temperature reached during the FJH process and the flash

healing effectiveness. Therefore, for these samples, the FJH voltage was set at 430 V and 410 V, respectively, resulting in F-LIG with a high and moderate decrease in resistance. Due to the limitations in the rising speed of power supply, the actual voltage is unable to reach the set value for voltages exceeding 200 V. As shown in Figure R8, when the voltage is set at 430 V, the actual voltage reached a maximum of 380 V. Consequently, the resulting P_A and Q_A reached approximately 1900 W cm⁻² and 25.72 J cm⁻², respectively. These values are comparable to those obtained from the 1 mm × 10 mm patterns treated under 190 V, suggesting a similar level of FJH process. At 410 V (actual voltage ~370 V), the P_A and Q_A reaches ~900 W cm⁻² and 14.21 J cm⁻², respectively, which is lower than the optimal case. Accordingly, the resistance decrease is moderate under this condition.

Figure R8. Curves of (a) voltage (green line), current (orange line), and (b) areal power density and integrated energy density for LIG patterns with dimensions of 1 mm × 20 mm.

Due to the significant amount of powder sample required for XRD testing, and considering that FJH technology is well-suited for powder samples, we employed powdered LIG for FJH treatment and subsequent XRD characterization. To carry out the FJH treatment, the powder sample was filled inside a quartz tube with an inner diameter of 10 mm. The fill thickness of the powder sample was approximately 8 mm. In this experimental setup, FJH was achieved by applying voltages of 70 V, 100 V, and 130 V. Compared with the LIG patterns, the Raman spectra of powder sample show a similar trend of variation. Figure R9 presents the specific I_D/I_G and corresponding L_a for the powder samples. As the applied voltage increases, the I_D/I_G decreases from 0.84 to 0.27, and the L_a increases from 23.0 to 71.9 nm. These parameters for the powdered LIG demonstrate a similar level of change as observed in the LIG patterns. The identical change degree in these parameters between the patterned and powdered LIG suggests that both samples undergo a comparable level of defect healing. The XRD analysis of the

powder sample points to the transformation of LIG from an amorphous to a crystalline phase, accompanied by a reduction in the interlayer spacing between the graphene sheets. This conclusion holds true to some extent for the pattern sample as well.

Figure R9. I_D/I_G and L_a of the LIG powder sample treated under different FJH voltages.

In the revised manuscript, we have included Figure R8 and R9 in the revised Supporting Information and the detailed discussions to clarify the use of different voltages for different samples. Thanks again for your suggestion to make our description clearer and more explicit.

6. *The application section of the article lacks innovation. (Liquid metal-based strain-sensing glove for human-machine interaction, 2023).*

Reply: We thank the reviewer's suggestion. A key contribution of our work is the development of a fabrication protocol for rapidly and effectively healing intrinsic defects in LIG, which is essential to improve the sensitivity of sensor. Thus, we mainly demonstrate the application pertaining to the defect healing. In this work, we show the advantages of the flash healing of LIG by using strain sensor as one demonstration. In the original manuscript, we have demonstrated that the F-LIG-based strain sensor exhibit significantly improved sensitivity in comparison to traditional LIG (Table R2 and this work). The high sensitivity makes it competent in detecting subtle human body motions, phonation recognition, and human-machine interfaces. Besides, due to the precise and real-time control of bending signals, it becomes possible to define the duration and frequency of signals to transmit Morse code for information encryption and communication. Figure R10 illustrates that by manipulating the endurance and frequency of finger bending, words such as "SOS" and "HELP" can be demonstrated.

Figure R10. (a) International Morse code and (b) Morse code for “SOS” and “HELP” produced by fingers bending.

In addition to strain sensors, we also explored the potential of F-LIG in low-voltage sterilization. To achieve this purpose, LIG films with dimensions of 10 mm × 10 mm were utilized. As mentioned in the original manuscript, through the FJH treatment, a notable decrease in resistance is observed from the initial value of ~250 Ω to 70 Ω. To quantitatively evaluate the antibacterial performance, a colony-forming unit (CFU) assay was conducted using *Escherichia coli* (*E. coli*) as model bacterium. The typical process involved the following steps: First, the LIG and F-LIG films were immersed into an *E. coli* suspension (10^8 CFU mL⁻¹) and incubated at 37 °C for 1 h. Next, the films were removed from the suspension and washed with ringer solution to remove unattached bacteria. Then, applying variable DC voltages (1 V, 3 V, and 5 V) to the films for 2 min. After that, the films were sonicated in 10 mL ringer solution for 5 min (10% power) to detach the bacteria. Bacteria CFU were enumerated using the plate counting method. Figure R11 and Figure R12 shows optical images of *E. coli* growth on agar plates and the corresponding CFU statistics and antibacterial efficiency for LIG and F-LIG films under different sterilization voltages. Initially, there is a similar adherence and survival of *E. coli* on both LIG and F-LIG, with a count of approximately 8.5×10^4 CFU mL⁻¹. Upon the application of low DC voltages, significant sterilization is observed. Notably, F-LIG exhibits a higher bacterial killing rate compared to LIG at each applied voltage. For instance, at 5 V, LIG shows a moderate bactericidal activity of 76.3%, while F-LIG achieves an excellent efficiency of 99.94%. Remarkably, the viable count of *E. coli* remains 2.09×10^4 CFU mL⁻¹ on LIG, whereas significantly reduced to only ~53 CFU mL⁻¹ on F-LIG.

Figure R11. Optical images of *E. coli* growth on agar plates for LIG and F-LIG films under different sterilization voltages.

Figure R12. (a) CFU statistics and (b) antibacterial efficiency for LIG and F-LIG films under different sterilization voltages.

During the process of electric sterilization, the pass-through current density and surface temperature were measured to investigate the electrothermal properties. Figure R13 demonstrates that due to the significantly enhanced conductivity of F-LIG, the current density in F-LIG is relatively higher than that in LIG under the same applied voltage. Consequently, F-LIG films are capable of achieving higher surface temperatures, and the heating was quite uniform, as evidenced by the infrared thermal images in Figure R14. For example, when the voltage is 5 V, the surface temperature of LIG reached only 33 °C, while F-LIG could reach temperatures above 57 °C. It has been reported that at moderate temperatures and short durations, electrical current is the primary factor responsible for bacterial killing, with the amount of bacterial eradication dependent on the current density (ACS Appl. Mater. Interfaces 2021,

13, 59373–59380). The mechanism of electrical sterilization is believed to involve the electroporation of bacteria and direct electron transfer between the bacteria and LIG.

Figure R13. Current densities passing through LIG and F-LIG under different sterilization voltages.

Figure R14. Infrared thermal images of LIG and F-LIG films during the electrical sterilization process.

In the revised manuscript, the above result and discussions have been incorporated. The detailed experimental procedures have been documented in the Method section. Furthermore, the revised manuscript now includes an outlook highlighting more promising applications of F-LIG, such as supercapacitors and chemical sensors. These potential applications will be further explored in our future work.

Reviewer: 3

Comments:

The manuscript of Cheng et al describes the formation of low resistance LIG structures by a simple flash joule heating process applied to fabricated LIG tracks and powders. This process sensibly reduces the resistance of

LIG by 5 times thus opening sensitive applications in health and electronics. The findings are very impressive and the manuscript well written, with relevant data provided to support findings and claims. the methodology is sound and the results are highly novel and interesting for the field.

I would recommend publication of the manuscript after addressing the following points:

Reply: We thank the reviewer's positive comments and suggestions for further improvement of the manuscript. Please see below our replies to comments.

1. It would be interesting to know if the flash healing step works with substrates that are different than polyimide or if it would be too harsh. although not mandatory, this extra information would open up applications in green electronics.

Reply: We thank the reviewer's suggestion. During the development of laser-induced graphene (LIG), various commercial polymers and natural materials have been employed as precursors. For example, aromatic polysulfone-class polymers, polysulfone (PSU) and polyethersulfone (PES), were utilized to produce sulfur-doped LIG with enhanced antibacterial and antifouling properties (*ACS Nano* 2018, 12, 289–297). Poly(ether-ether-ketone) (PEEK), a versatile engineering plastic, has shown promise in producing LIG with diverse applications such as humidity sensors and supercapacitors (*ACS Appl. Nano Mater.* 2023, 6, 19, 17802–17813; *J Mater. Sci.* 2019, 54, 4192–4201). Lignin, a widely used biomass material, has also found considerable use in LIG electronics production owing to its sustainability, cost-effectiveness, and eco-friendly characteristics.

According to your suggestion, in addition to the commonly used polyimide (PI), we further developed experiments using different substrates, including PES, PEEK, and PES/lignin, to explore the adaptability and universality of the flash healing process to LIG patterns. PEEK film with thickness of 250 μm was provided by Meideyuan Plastic Products Co., LTD, China. The PES and PES/lignin films were fabricated using solution casting method. In a typical procedure, 1.2 g of PES commercial polymer powders were dissolved in 10 mL dimethylformamide (DMF). The resulting homogenous solution was poured into an aluminum dish (inner diameter 6 cm) and kept at 80 °C overnight to obtain PES films. To prepare PES/lignin film, an additional 0.8 g of lignin was added to the PES solution, and the casting was carried out in the same manner. Subsequently, dumbbell-shaped LIG patterns were engraved onto these polymer films and then subjected to 20 ms DC voltages. Figure R15 illustrates the successful triggering of the FJH process, evident from the observation of bright flash. Figure R16 demonstrates

that, regardless of the substrate used, the resulting F-LIG exhibit reduced resistance and improved Raman signals.

Due to the lower thermal stability of these substrates compared to PI, we could only apply lower FJH energy densities. Thus, the resistance decrease ratio for these three kinds of LIG patterns is not as high as that for PI-LIG. Nonetheless, the extension of the flash healing method to LIG with different substrates remains highly significant, demonstrating considerable potential of this technique in a broader range of applications.

Figure R15. FJH process of PEEK-, PES-, and PES/lignin- derived LIG patterns.

Figure R16. Resistance and Raman spectra of PEEK-, PES-, and PES/lignin- derived LIG patterns before and after FJH process.

Figure R15 and R16 have been added to the revised Supplementary Information. The above result and discussions were also added to the revised manuscript.

2. Although the performance of strain sensor and other sensors shown in Fig 5 are greatly improved compared to LIG, it would be good to add a comparative table (may even in the SI).

Reply: Thank you for the advice. We conducted a comprehensive literature survey and compared the sensing sensitivity of our F-LIG-based strain sensors with recently reported piezoresistive strain sensors,

with a specific focus on low strain ranges. As listed in Table R2, the sensitivity of our F-LIG/PDMS strain sensor stands out among state-of-the-art LIG-based sensors and surpasses many metal- and other 1D or 2D nanomaterials-based sensors.

Table R2. Sensitivity comparison of state-of-the-art strain sensors under low strain ranges

Sensing material	GF/strain ranges	Reference
LIG/PDMS	38 (0-10%)	ACS Appl. Mater. Interfaces 2022, 14, 36, 41283–41295
LIG/PU	40 (0-10%)	ACS Appl. Mater. Interfaces 2020, 12, 19855 –19865
Lignin-LIG/PDMS	20 (0-60%)	Sens. Actuators, A 2022, 334, 113320
Wavy-LIG/PDMS	37.8 (0-32%)	Sensors 2020, 20, 4266
HfSe ₂ /LIG-Ecoflex	46 (0-30%)	Appl. Surf. Sci. 2023, 636, 157772
BP/LEG on SEBS	81 (0-7.5%)	Adv. Funct. Mater. 2021, 31, 2007661
MWCNTs/graphene/silicone rubber/Fe ₃ O ₄ nanocomposites	8.43 (0-120%)	Small 2023, 19, 2205316
Ti ₃ C ₂ T _x MXene/CNT	64.6 (0-30%)	ACS Nano 2018, 12, 1, 56–62
Ti ₃ C ₂ T _x MXene/AgNWs/LM-PDMS	3.22 (0-100%)	Adv. Funct. Mater. 2023, 33, 2301587
CNT-PDMS	14.5 (0-20%)	Adv. Funct. Mater. 2021, 31, 2103375
F-LIG-H/PDMS	120 (2-10%)	This work

*PU-polyurethane; BP-black phosphorus; LEG-laser-engraved graphene; MWCNTs-multi-walled carbon nanotubes; CNT-carbon nanofiber; AgNWs-Ag nanowires; LM-liquid metal

We have added table R2 in the revised Supplementary Information as Table S1 and added the following sentences in the revised manuscript: “Our sensor exhibits relatively high GF among state-of-the-art LIG-based sensors and outperforms many metal- and other low-dimensional nanomaterials-based sensors, highlighting its competence in detecting subtle deformations with exceptional accuracy”.

REVIEWERS' COMMENTS

Reviewer #1 (Remarks to the Author):

The authors thoroughly reviewed the paper and addressed all the points raised by me and other reviewers. They not only answered very well to those questions but also added significant amount of data and rationalization, improving very much the quality, interest and content of the paper. Most importantly, by considering the positive comments and suggestions, they could provide much more insight in the mechanisms governing the FJH and its effects on LIG. This paper now contains information which will be very useful for the scientific community. I suggest to accept the manuscript for publication.

Reviewer #2 (Remarks to the Author):

The manuscript can be accepted now

Reviewer #3 (Remarks to the Author):

I am satisfied that the authors have addressed all previously raised points and I recommend the publication of the article as is